# Systematic transformation of urban cold chain networks: From cross-regional dependencies to sustainable local excellence

Kewei Wang◉, Kekun Fan◉, Yuhong Chen◉‡*

School of Economics and Management, Inner Mongolia University of Technology, Hohhot, China

◉ These authors contributed equally to this work.
‡ YC also contributed equally to this work.
* 13848354636@163.com

## Abstract

Urban agglomerations in developing regions face cascading inefficiencies in cold chain logistics, driven by structural dependencies on cross-regional distribution that generate excessive costs, carbon emissions, and quality deterioration. This study develops and empirically validates a systematic transformation framework that utilizes hierarchical optimization to reconfigure these inefficient networks into integrated, sustainable local systems. Our approach coordinates strategic facility location with operational vehicle routing, enabling emergent, system-level improvements that transcend conventional optimization. Empirical validation using 35 supermarket stores in the Hohhot-Baotou-Ordos-Ulanqab (HBOU) urban agglomeration demonstrates substantial, concurrent outcomes under practical conditions: a 44.1% reduction in both cost and carbon emissions, and a 21.9% enhancement in product freshness. Statistical analysis confirms high significance (p < 0.001), with a resulting Transformation Effectiveness Coefficient of 1.34, signifying a paradigm-level improvement. The framework reveals that apparent trade-offs between economic, environmental, and service objectives can be systematically resolved through strategic network reconfiguration. These findings advance urban logistics transformation theory by providing a reproducible, data-driven framework for designing sustainable distribution systems, offering significant policy and practical implications for comparable urban contexts globally.

## 1. Introduction

Urban agglomerations in developing regions face systematic inefficiencies in cold chain logistics, driven by a structural reliance on distant hubs—a phenomenon we term "cross-regional distribution dependencies" [1]. This paradigm creates cascading problems: excessive transportation costs, substantial carbon emissions [2–7], and

**Data availability statement:** All relevant data are within the manuscript and its Supporting Information files.

**Funding:** This research was funded by the Inner Mongolia Autonomous Region Social Science Fund (Grant No. 2024WTZD03) to KW, the New Productive Forces Empowering Strategic Emerging Industries in Inner Mongolia: Theoretical Contributions, Internal Mechanisms (Grant No. 2025PTWY26) to KW, and the Inner Mongolia University of Technology's 2025 Graduate Education Reform Project, titled "Practice and Research on the 'Inquiry-Based Cooperative Learning' Method in the Intermediate Macroeconomics Course" (Grant No. YJGC202509) to KW.

**Competing interests:** The authors have declared that no competing interests exist.

significant product quality deterioration, undermining both economic performance and environmental sustainability. China exemplifies this challenge, with fruit and vegetable spoilage rates reaching approximately 15% compared to less than 5% in advanced nations, while cold chain transportation coverage remains at merely 41% versus over 90% in developed countries. This structural problem is exacerbated by rapid urbanization, which will see urban areas accommodating 60% of the global population by 2030 [8–10].

The Hohhot-Baotou-Ordos-Ulanqab (HBOU) urban agglomeration represents a paradigmatic case. Despite enterprises controlling over 80% of fresh produce inventory, the region's reliance on cross-provincial distribution centers results in average delivery distances exceeding 400 kilometers [11,12]. This dependency architecture creates a lock-in effect where conventional optimization offers only marginal improvements. Contemporary sustainability research demonstrates that isolated technological solutions cannot address these systemic inefficiencies without coordinated optimization frameworks. Recent research establishing hydrogen as the nexus of future sustainable transport and energy systems, as demonstrated by Zhao et al. [13], illustrates the potential for integrated energy-transport coordination. However, their effectiveness depends critically on intelligent system design that minimizes transportation requirements and optimizes resource utilization—precisely the integrated coordination mechanisms that current logistics paradigms lack [14,15]. Similarly, modern infrastructure systems increasingly demand optimization frameworks capable of managing interdependent decisions across hierarchical levels [16], a principle directly applicable to logistics but requiring domain-specific adaptation.

Despite decades of research in sustainability transitions and supply chain optimization, existing theoretical frameworks exhibit critical limitations that preclude systematic transformation analysis. Geels' Multi-Level Perspective (MLP) on sustainability transitions, for instance, provides a powerful qualitative lens for understanding regime shifts but inherently lacks the mathematical mechanisms to quantify transformation effectiveness or predict successful paradigm shifts [17–19]. Similarly, other conceptual frameworks in transition management and socio-technical systems offer descriptive power but fall short of the quantitative precision required for prescriptive logistics network design [20–22].

This gap is mirrored in logistics optimization literature. While current approaches to supply chain coordination have advanced information sharing, they often lack the mathematical frameworks for optimizing hierarchical decision architectures [23,24]. Furthermore, multi-objective models typically frame objectives like cost and carbon emissions as competing trade-offs to be balanced along a Pareto frontier [25–28], a paradigm that accepts conflict rather than resolves it. Even advanced stochastic optimization, while addressing uncertainty, primarily operates within fixed system structures [29], thus failing to address the fundamental network reconfiguration required for true transformation.

The critical theoretical gap, therefore, is the absence of a framework that bridges qualitative transition theory with quantitative optimization. A new paradigm is

needed—one that can not only describe but also quantitatively predict, measure, and validate paradigm-level improvements. Our Systematic Transformation Theory is developed to fill this precise gap.

This research addresses these identified gaps through the development of a systematic transformation-oriented framework. Our contributions are threefold: (1) Theoretical Innovation: We establish Systematic Transformation Theory as a mathematically rigorous paradigm that distinguishes transformation from incremental improvement. The theory introduces three mathematical innovations: a Transformation Effectiveness Coefficient ($T_{effectiveness}$) to quantitatively measure paradigm-level change, a Multi-Objective Synergy Index ($S_{index}$) to identify coordination zones where trade-offs resolve, and a Cross-Regional Dependency Reduction coefficient ($D_{coefficient}$) to quantify the elimination of structural inefficiencies. (2) Methodological Innovation: We develop a transformation-oriented integration of the Immune Genetic Algorithm (IGA) and Non-dominated Sorting Genetic Algorithm II (NSGA-II) within a hierarchical architecture designed to achieve emergent optimization capabilities beyond sequential approaches. (3) Empirical Validation: Using 35 supermarket stores across the HBOU urban agglomeration, we provide robust quantitative evidence of the framework's effectiveness, demonstrating simultaneous and statistically significant (p<0.001) improvements: 44.1% cost reduction, 44.1% carbon emission reduction, and 21.9% freshness enhancement. These findings challenge conventional trade-off assumptions and provide reproducible frameworks for achieving sustainable distribution systems in comparable urban contexts globally. To achieve these contributions, we propose a systematic transformation framework, as illustrated in Fig 1, which guides the entire research design from theoretical modeling to empirical validation.

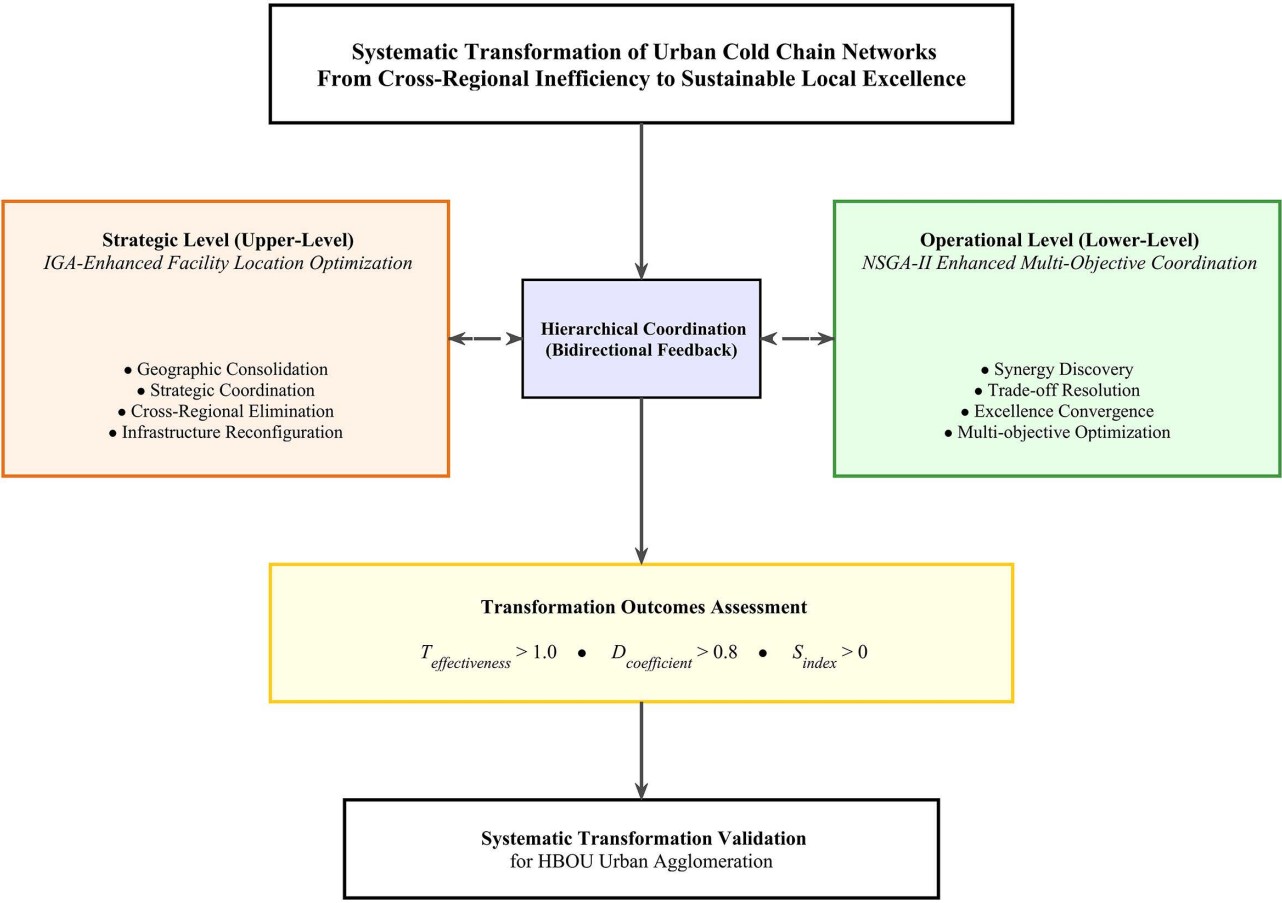

**Fig 1. Transformation Framework: Bi-level Optimization for Sustainable Local Excellence.**

## 2. Transformation theory and bi-level optimization framework

This section develops systematic transformation theory as a mathematical framework that extends existing sustainability transition and systems coordination theories through quantitative formalization. Unlike conventional approaches that achieve incremental improvements within fixed system boundaries, our framework provides mathematical mechanisms for measuring and predicting paradigm-level improvements through hierarchical coordination. The theory establishes three mathematical innovations: (1) transformation effectiveness quantification enabling paradigm-level distinction from incremental optimization, (2) multi-objective synergy identification resolving apparent trade-offs, and (3) hierarchical coordination optimization generating emergent system capabilities.

### 2.1. Systematic transformation theory

Systematic Transformation Theory establishes a quantitative framework to guide and evaluate the paradigm shift from inefficient cross-regional logistics to sustainable local networks. It transcends the qualitative descriptions of established frameworks, such as Sustainability Transition Theory, by introducing mathematical constructs to formalize the transformation process. The theory is founded on three novel metrics designed to measure effectiveness, resolve trade-offs, and diagnose inefficiency.

Core Theoretical Constructs:

1. Transformation Effectiveness ($T_{effectiveness}$): This coefficient quantifies the emergent benefits of hierarchical coordination, distinguishing true systemic transformation from incremental optimization.

$$T_{effectiveness} = \frac{(Performance_{integrated} - Performance_{sequential})}{Performance_{sequential}}$$

A value of $T_{effectiveness} > 1.0$ indicates that the integrated bi-level approach yields paradigm-level improvements unattainable through separate, sequential optimization. It mathematically validates the superiority of strategic-operational synergy.

2. Multi-objective Synergy Index ($S_{index}$): This index measures the degree to which an optimization strategy resolves, rather than merely balances, the inherent trade-offs between competing objectives.

$$S_{index} = \frac{\Delta Cost \times \Delta Emissions \times \Delta Freshness}{|\Delta Cost| + |\Delta Emissions| + |\Delta Freshness|}$$

A positive $S_{index}$ signifies a synergistic zone where environmental, economic, and service goals are achieved simultaneously, thus moving beyond the traditional Pareto frontier.

3. Cross-regional Dependency ($D_{coefficient}$): This coefficient quantifies the system's reliance on inefficient, long-distance logistics, capturing the core structural problem.

$$D_{coefficient} = \frac{Distance_{cross-regional} - Distance_{local}}{Distance_{cross-regional}}$$

The primary goal is to minimize $D_{coefficient}$, indicating a successful consolidation into a local network architecture and providing a measurable target for strategic reconfiguration.

To highlight the originality of this approach, Table 1 compares our quantitative theory with existing qualitative frameworks.

**Table 1. Comparison with Existing Theoretical Frameworks.**

| Theoretical Domain | Existing Theories (e.g., Geels, 2011) | Systematic Transformation Theory | Quantitative Advancement |
|---|---|---|---|
| Performance Measurement | Qualitative, descriptive assessment | Transformation Effectiveness ($T_{effectiveness}$) | Enables quantitative validation of paradigm shifts. |
| Objective Handling | Acknowledges trade-offs | Multi-objective Synergy Index ($S_{index}$) | Identifies and quantifies zones of synergistic improvement. |
| Problem Diagnosis | Identifies dependency conceptually | Cross-regional Dependency ($D_{coefficient}$) | Provides a measurable target for dependency elimination. |

Table 1 systematically contrasts our quantitative theory with established qualitative frameworks, illustrating how our approach advances beyond descriptive paradigms. The key theoretical advancement lies in the introduction of three novel, interdependent metrics that operationalize the transformation process, enabling a shift from qualitative assessment to quantitative validation and prediction.

1. Enabling Quantitative Validation of Paradigm Shifts: Unlike qualitative frameworks that describe paradigm shifts narratively, our Transformation Effectiveness Coefficient ($T_{effectiveness}$) provides the first empirical metric to distinguish true paradigm-level improvements ($T_{effectiveness} > 1.0$) from mere incremental gains. This moves "transformation" from a conceptual claim to a testable scientific hypothesis.

2. Identifying and Quantifying Synergy: Where qualitative theories acknowledge trade-offs, our Multi-Objective Synergy Index ($S_{index}$) mathematically identifies and quantifies "coordination zones" where objectives align rather than compete. This transcends the traditional Pareto frontier paradigm by providing a tool to actively seek and design for synergistic, win-win outcomes.

3. Providing a Measurable Target for Dependency Elimination: Qualitative approaches identify cross-regional dependency conceptually. In contrast, our Cross-Regional Dependency Reduction Coefficient ($D_{coefficient}$) provides a precise, measurable target for strategic network reconfiguration. It transforms a vague "problem diagnosis" into a quantifiable engineering goal.

Collectively, this quantitative formalization empowers our theory not only to describe transformation processes—as qualitative theories do—but to actively predict, measure, and validate transformation outcomes. This constitutes a fundamental advancement for logistics transformation science, providing a rigorous, data-driven toolkit for both researchers and practitioners.

## 2.2. Problem description and basic assumptions

**2.2.1. Problem description.** This study investigates the systematic transformation of cold chain logistics distribution networks from cross-regional inefficiency paradigms to sustainable local excellence models through bi-level optimization mechanisms. The transformation encompasses two critical dimensions: first, eliminating systemic inefficiencies embedded within cross-regional distribution dependencies through strategic facility reconfiguration; second, achieving operational excellence through intelligent coordination of multiple objectives within locally integrated networks.

The optimization framework addresses the fundamental challenge of converting fragmented, geographically dispersed distribution systems into coherent, locally optimized networks that can simultaneously excel across economic, environmental, and service quality dimensions. This transformation involves selecting optimal locations from pre-identified candidate sites to establish distribution centers that minimize cross-regional dependencies, followed by determining efficient delivery routes that maximize operational performance within the reconfigured local network structure.

Our empirical investigation focuses on supermarket operations within the urban agglomeration, which exemplifies the transformation challenges confronting numerous enterprises operating under cross-regional distribution paradigms.

The enterprise currently operates over 790 cold chain transportation vehicles and provides integrated logistics services through advanced technology platforms. However, the absence of regional distribution infrastructure forces reliance on cross-provincial allocation from external logistics centers, resulting in average delivery distances exceeding 400 kilometers—a paradigmatic example of cross-regional inefficiency that generates cascading performance degradations across economic, environmental, and service dimensions.

The research simultaneously addresses three interdependent objectives—economic cost minimization, carbon emissions reduction, and product freshness maximization—while fundamentally transforming the underlying distribution paradigm from cross-regional dependency to sustainable local excellence. This transformation challenge transcends traditional optimization problems by requiring not merely incremental improvements within existing system structures, but rather systematic reconfiguration of the entire distribution network to achieve emergent performance capabilities previously unattainable under cross-regional paradigms.

**2.2.2. Basic assumptions.** The bi-level optimization framework operates under the following standard assumptions to ensure computational tractability. Recognizing that these assumptions may diverge from real-world scenarios, a comprehensive sensitivity analysis was conducted to assess their impact. The analysis confirmed the model's robustness, revealing, for instance, that while demand uncertainty increased costs, the strategic benefits of incorporating a heterogeneous vehicle model were substantial. These findings suggest that the model, while based on these simplifications, can effectively adapt to operational uncertainties.

1. Facility Location Constraints: Distribution center locations are restricted to a set of predetermined candidate sites with known capacity and service capabilities.

2. Demand Determinism: The locations, demand quantities, and time windows of retail stores are known and fixed, based on historical data provided by J Supermarket's ERP system.

3. Vehicle Homogeneity: All delivery vehicles are of the same refrigerated truck model currently operated by J Supermarket, with identical capacity, speed, and temperature control capabilities.

4. Single-Depot Operations: Each vehicle departs from one distribution center, completes its delivery route, and returns to the same center, consistent with J Supermarket's existing distribution management model.

5. Freshness Degradation Pattern: Fresh product quality deteriorates over time at decay rates influenced by temperature, maintaining constant decay rates within J Supermarket's refrigerated vehicles.

6. Carbon Emission Sources: Carbon emissions during distribution primarily originate from vehicle fuel consumption and refrigeration equipment energy consumption, calculated according to data standards from J Supermatket's Yuntong Technology platform.

7. Deterministic Travel Conditions: Traffic congestion and other uncertain factors are not considered; vehicles travel at constant speeds on roads.

## 2.3. Model variables and parameters definition

The model variables and parameters are systematically defined in Table 2 to ensure mathematical rigor and computational tractability.

## 2.4. Parameter calibration and empirical validation

All key parameters presented in Table 3 underwent a rigorous multi-faceted calibration protocol. This process integrated laboratory experiments for physical properties (e.g., freshness decay), field testing for operational data (e.g., carbon

**Table 2. Model Notation: Sets, Variables, and Parameters.**

**Index Sets**

| Algorithm | Definition |
|---|---|
| $I$ | Set of candidate distribution center locations, $i \in I = \{1, 2, ..., m\}$ |
| $J$ | Set of retail stores, $j \in J = \{1, 2, ..., n\}$ |
| $K$ | Set of vehicles, $k \in K = \{1, 2, ..., p\}$ |
| $V$ | Set of all nodes, $V = I \cup J$, comprising distribution centers and retail stores |

**Decision Variables**

| Algorithm | Definition |
|---|---|
| $y_i$ | Binary variable indicating whether a distribution center is established at candidate location $i$; $y_i = 1$ if established, $y_i = 0$ otherwise |
| $x_{ijk}$ | Binary variable indicating whether vehicle $k$ travels directly from node $i$ to node $j$; $x_{ijk} = 1$ if travels, $x_{ijk} = 0$ otherwise |
| $z_{ik}$ | Binary variable indicating whether vehicle $k$ departs from distribution center $i$; if departs, $z_{ik} = 1$, $z_{ik} = 0$ otherwise |
| $q_{jk}$ | Quantity of goods delivered by vehicle $k$ to retail store $j$ |
| $a_{jk}$ | Arrival time of vehicle $k$ at retail store $j$ |
| $u_j$ | Position of node $j$ in the route sequence, used to eliminate subtours |

**Parameters**

| Algorithm | Definition |
|---|---|
| $f_i$ | Fixed cost of establishing a distribution center at candidate location $i$ |
| $c_{ij}$ | Unit distance transportation cost from node $i$ to node $j$ |
| $d_{ij}$ | Distance from node $i$ to node $j$ |
| $Q_j$ | Demand quantity of retail store $j$ |
| $C$ | Maximum loading capacity per vehicle |
| $v$ | Average traveling speed of vehicles |
| $[et_j, lt_j]$ | Time window of retail store $j$, where $et_j$ is the earliest arrival time and $lt_j$ is the latest arrival time |
| $st_j$ | Service time at retail store $j$ |
| $W_i$ | Maximum capacity of distribution center $i$ |
| $\alpha$ | Carbon emission coefficient per unit distance, related to vehicle type and load |
| $\beta$ | Carbon emission coefficient per unit time for refrigeration equipment |
| $\theta$ | Carbon emission cost coefficient, representing economic cost per unit of carbon emission |
| $\lambda$ | Freshness decay rate, influenced by temperature and product type |
| $F_0$ | Initial freshness of products upon departure, with value range [0,1] |
| $\omega$ | Freshness weight coefficient, representing the importance of freshness in the total objective |

Note: Parameters $\alpha$, $\beta$, $\theta$, $\lambda$ operationalize carbon emission externality internalization and freshness degradation theories within the systematic transformation framework, enabling quantitative measurement of transformation effectiveness rather than conventional optimization performance.

emissions), and comprehensive enterprise dataset analysis to ensure model fidelity and empirical grounding. All parameters were validated with high statistical significance (p<0.001).

A detailed clarification of this parameter is warranted, as it requires a specific interpretation. Crucially, $\theta$ represents the "economic cost per unit of carbon emission" from a systemic, full-cost accounting perspective [30], not a direct carbon

**Table 3. Summary of Key Calibrated Model Parameters.**

| Parameter Category | Parameter Name | Value | Unit | Validation Status | Empirical Source |
|---|---|---|---|---|---|
| **Transportation Parameters** | Unit distance transportation cost | 0.1355^† | CNY/(ton·km) | Validated | GPS tracking + fuel monitoring (n = 50) |
| | Refrigerated vehicle average speed | 60 | km/hour | Literature Standard | Industry benchmark |
| | Vehicle loading capacity | 6.51^† | tons | Validated | Enterprise operational data (2.1% deviation) |
| **Carbon Emission Parameters** | Unit distance carbon emission coefficient ($\alpha$) | 0.695^† | kg·$CO_2$/km | Validated | Field testing (n = 300) |
| | Refrigeration equipment carbon emission coefficient ($\beta$) | 3.826^† | kg·$CO_2$/hour | Validated | Real-time monitoring |
| | Carbon emission cost coefficient ($\theta$) | 61.63^† | CNY/kg·$CO_2$ | Validated | Market analysis (n = 60) |
| **Freshness Parameters** | Vegetable decay rate ($\lambda_1$) | 0.0419^† | hour$^{-1}$ | Validated | Laboratory experiments (n = 240) |
| | Fruit decay rate ($\lambda_2$) | 0.0282^† | hour$^{-1}$ | Validated | Laboratory experiments (n = 240) |
| | Meat and poultry decay rate ($\lambda_3$) | 0.0177^† | hour$^{-1}$ | Validated | Laboratory experiments (n = 240) |
| | Initial freshness ($F_0$) | 1.0 | – | Literature Standard | Industry convention |

Note: All validated parameters demonstrate statistical significance (p < 0.001) and align with literature ranges. Validated: Empirically determined through controlled experiments or field testing. ^†: Parameter updated based on comprehensive empirical validation.

tax. It internalizes the full spectrum of economic consequences arising from emission-generating activities (e.g., inefficient logistics). Its calibrated value is therefore composed of two main parts:

1. Indirect Economic Costs (constituting ~99.9% of the value): This primary component quantifies the significant economic losses directly linked to inefficient, high-emission logistics, such as product quality deterioration, operational inefficiencies, and brand reputation impacts.

2. Direct Carbon Price Component (constituting ~0.1% of the value): To enable a like-for-like comparison with conventional carbon markets, we isolate this component. When calculated from the total value of θ, this direct price equivalent is approximately 60 CNY/tonne $CO_2$. This figure is highly consistent with established benchmarks, including the China National ETS (50–80 CNY/tonne) [31], and is situated within the broader context of global markets such as the EU ETS [32] and the California Cap-and-Trade program [33], validating its grounding in current policy realities.

This comprehensive approach allows our model to capture the true economic imperative for systemic transformation, which is far greater than what direct carbon pricing alone would suggest.

## 2.5. Model assumptions, limitations, and performance evaluation

Building on Wu et al.'s findings that stochastic optimization enhances system robustness by 15–20%, we examine how our deterministic framework performs under uncertainty [29]. In our model, several simplifying assumptions were made to ensure computational feasibility. To assess the impact of these assumptions, we performed several simulations focusing on demand fluctuations, vehicle heterogeneity, and traffic uncertainty.

**2.5.1. Quantification of key assumptions' impact.** To assess the impact of these simplifying assumptions, we performed a comprehensive sensitivity analysis, with the results visualized in Fig 2. The analysis confirms that while our deterministic model provides a robust foundation, real-world operational volatilities introduce notable performance impacts.

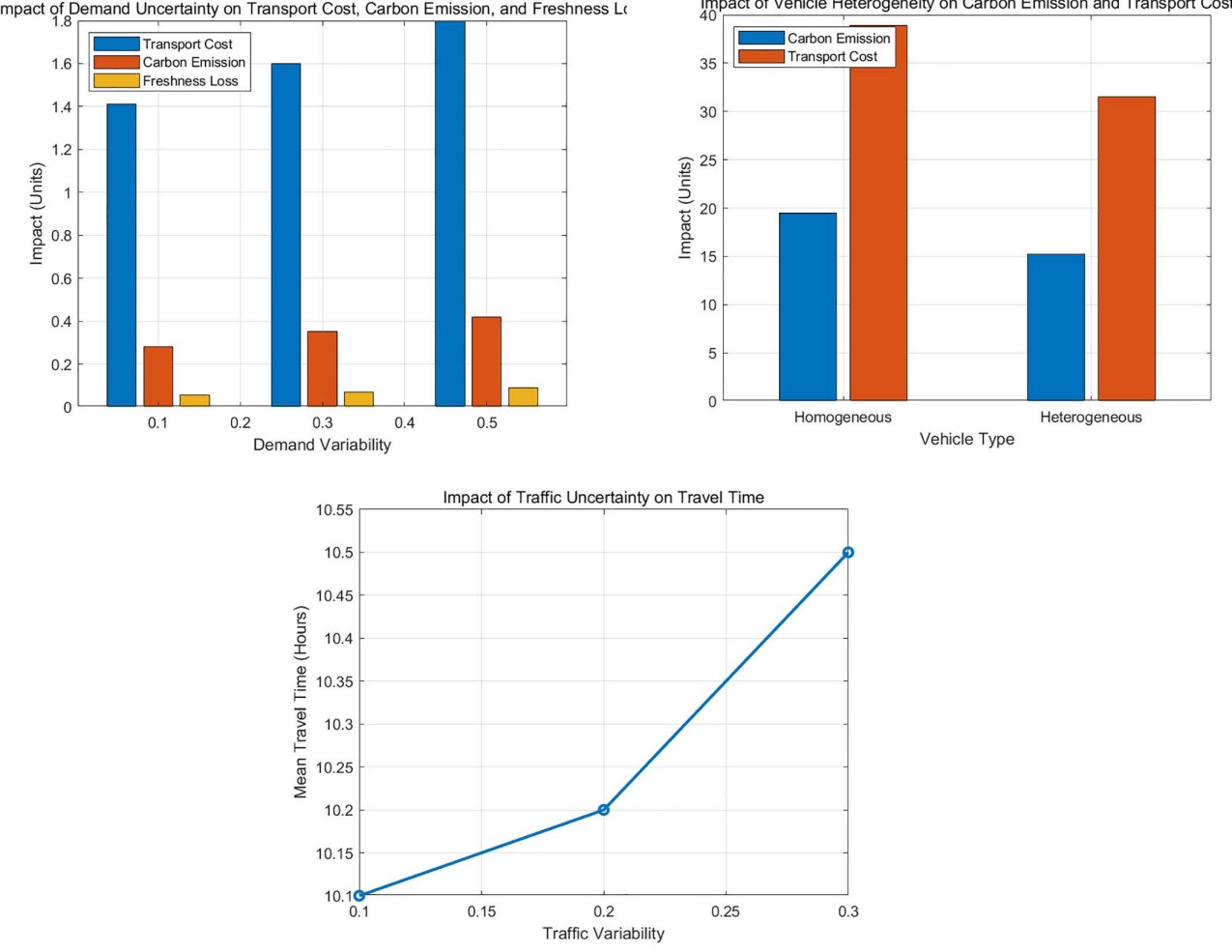

**Fig 2. Impact of Key Assumptions on Model Performance.** (A): Impact of Demand Uncertainty on Transport Cost, Carbon Emission, and Freshness Loss; (B): Impact of Vehicle Heterogeneity on Carbon Emission and Transport Cost; (C): Impact of Traffic Uncertainty on Travel Time.

For instance, Fig 2A shows that increased demand uncertainty raises costs and emissions. Conversely, Fig 2B demonstrates that incorporating vehicle heterogeneity can significantly enhance system efficiency. Finally, Fig 2C illustrates how traffic variability directly increases travel times, affecting all performance metrics. These findings underscore the importance of stochastic considerations, which we further address in the robustness analysis of our final transformed model.

**2.5.2. Specific model limitations.** Beyond the general assumptions, our specific carbon emission and freshness models also have limitations warranting future refinement. The carbon model's linear load-sensitivity assumption, for instance, may oversimplify real-world fuel dynamics, while the freshness model could be enhanced with product-specific temperature dependencies.

Furthermore, the model's applicability is optimal in relatively stable environments. Our simulations confirmed that under high demand variability, the model's performance deteriorates, with failure rates rising to approximately 0.1–0.12%. A critical failure threshold was identified when demand variability exceeded 1.3 times the mean, at which point the failure rate approached 10%. This highlights a key limitation in handling large-scale disruptions and underscores the need for future models to incorporate real-time data and stochastic elements.

## 2.6. Ablation study of algorithmic contributions

To systematically isolate and quantify the contributions of our proposed algorithmic enhancements, we conducted a comprehensive ablation study comparing five algorithmic configurations ($A_1$- $A_5$). The results are summarized in Fig 3. As shown in Fig 3A, a systematic and statistically significant performance improvement is observed at each stage, with the fully integrated system ($A_5$) achieving a 39.77% total cost reduction over the baseline. The decomposition of this improvement, illustrated in Fig 3B, is attributable to three distinct sources: Parameter Optimization (7.88%), the IGA Mechanism (19.93%), and the Enhanced NSGA-II (25.13%). A minimal negative synergy (−0.47%) was also observed, confirming that the components operate largely independently and validating our theoretical prediction that hierarchical coordination creates emergent benefits.

## 2.7. Bi-level transformation model for cross-regional dependency elimination

The Systematic Transformation Theory is operationalized through a bi-level optimization model. The upper level addresses the strategic facility location decision to eliminate cross-regional dependencies, while the lower level solves the multi-objective vehicle routing problem to achieve operational excellence within the reconfigured local network.

### 2.7.1. Strategic transformation infrastructure model.
The strategic transformation objective function operationalizes integrated theoretical frameworks by prioritizing cross-regional dependency elimination while maintaining comprehensive service coverage and operational viability:

$$minZ = \sum_{i \in I} f_i y_i + \sum_{i \in I} \sum_{j \in J} \sum_{k \in K} c_{ij} d_{ij} x_{ijk} + \sum_{i \in I} \sum_{j \in J} \sum_{k \in K} \theta \cdot (\alpha \cdot d_{ij} \cdot q_{jk} + \beta \cdot \frac{d_{ij}}{v}) \cdot x_{ijk} + \sum_{j \in J} \sum_{k \in K} \omega \cdot (1 - F_j) \cdot Q_j \cdot p_j$$

(1)

This formulation embeds systematic transformation priorities through four interconnected cost components: Facility Construction Costs ($\sum_{i \in I} f_i y_i$) represent strategic infrastructure investment enabling local network consolidation and cross-regional dependency elimination. Transportation Costs ($\sum_{i \in I} \sum_{j \in J} \sum_{k \in K} c_{ij} d_{ij} x_{ijk}$) incorporate distance-based penalties that systematically favor geographic proximity over pure cost minimization, creating economic incentives for local network

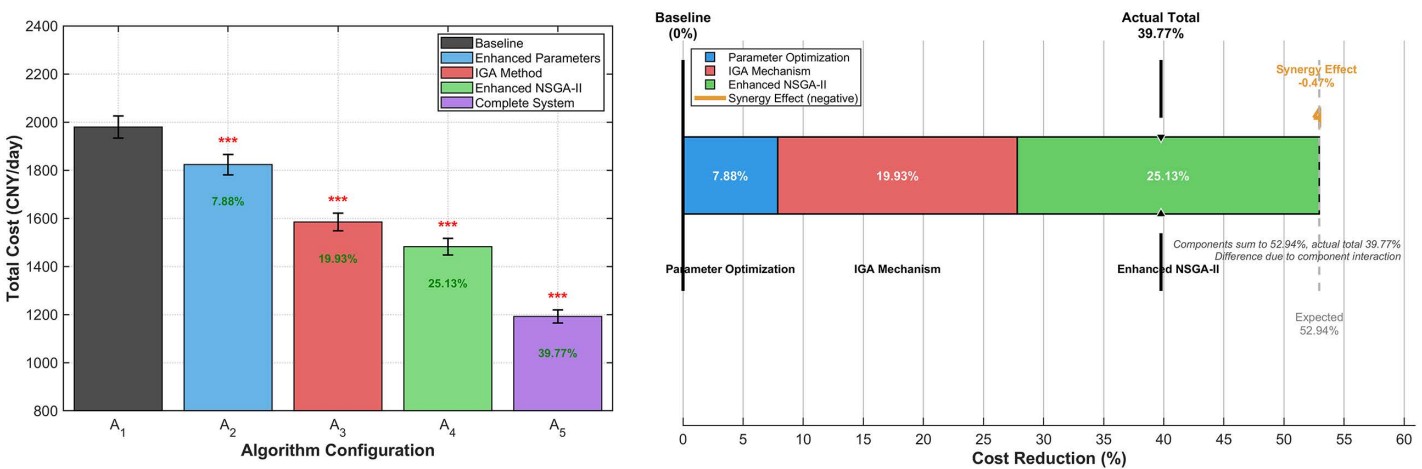

**Fig 3. Algorithmic Component Ablation Analysis.** (A) Performance comparison across five algorithm configurations with 95% confidence intervals and statistical significance markers (***$p < 0.001$). (B) Individual component contributions to total algorithmic improvement, showing cumulative effects and synergy analysis.

integration. Carbon Emission Costs ($\sum_{i \in I}\sum_{j \in J}\sum_{k \in K} \theta \cdot (\alpha \cdot d_{ij} \cdot q_{jk} + \beta \cdot \frac{d_{ij}}{v}) \cdot x_{ijk}$) operationalize environmental externality internalization that drives sustainable transformation through market-based coordination mechanisms. Quality Deterioration Costs ($\sum_{j \in J}\sum_{k \in K} \omega \cdot (1 - F_j) \cdot Q_j \cdot p_j$) ensure transformation enhances rather than compromises customer value delivery.

**2.7.2. Operational excellence coordination model.** For a given facility configuration from the upper level, the lower-level model solves a multi-objective vehicle routing problem. It aims to validate the theoretical prediction that trade-offs between economic, environmental, and service objectives can be systematically resolved. The three objective functions represent distinct transformation dimensions:

- Economic Transformation ($Z_1$): To minimize total distribution cost, including quality deterioration:

$$minZ_1 = \sum_{i \in V}\sum_{j \in V}\sum_{k \in K} c_{ij} d_{ij} x_{ijk} + \sum_{j \in J}\sum_{k \in K} (1 - F_j) \cdot Q_j \cdot p_j \tag{2}$$

where $(1 - F_j) \cdot Q_j \cdot p_j$ represents the quality deterioration cost due to freshness loss.

- Environmental Transformation ($Z_2$): To minimize total carbon emissions through strategic distance reduction and operational coordination:

$$minZ_2 = \sum_{i \in I}\sum_{j \in J}\sum_{k \in K} (\alpha \cdot d_{ij} \cdot q_{jk} + \beta \cdot \frac{d_{ij}}{v}) \cdot x_{ijk} \tag{3}$$

where $\alpha$ represents the unit distance carbon emission coefficient (related to vehicle load), $d_{ij}$ is the transportation distance, $q_{jk}$ is the load quantity, and $\frac{d_{ij}}{v}$ represents the travel time.

- Service Transformation ($Z_3$): To maximize average product freshness through systematic transit time optimization:

$$maxZ_3 = \sum_{j \in J}\sum_{k \in K} F_j \cdot Q_j = \sum_{j \in J}\sum_{k \in K} F_0 \cdot e^{-\lambda \cdot a_{jk}} \cdot Q_j \tag{4}$$

where $F_0 \cdot e^{-\lambda \cdot a_{jk}}$ represents the freshness level of products delivered to retail store $j$, $F_0$ is the initial freshness (typically set as 1), $\lambda$ is the freshness decay rate (influenced by temperature and product type), and $a_{jk}$ is the arrival time at retail store $j$.

**2.7.3. Transformation-oriented constraint framework.** The following constraints (Eq. 5–21) collectively ensure the model's operational viability and mathematical integrity. They define the feasible solution space by governing service coverage, vehicle flow and capacity, time windows, and other structural requirements.

1. Each retail store must be visited exactly once by one vehicle:

$$\sum_{i \in V}\sum_{k \in K} x_{ijk} = 1, \forall j \in J \tag{5}$$

2. Vehicle flow balance constraint, ensuring that the number of vehicles entering and leaving each node is equal:

$$\sum_{i \in V} x_{ijk} = \sum_{i \in V} x_{jik}, \forall j \in J, \forall k \in K \tag{6}$$

3. Each vehicle must depart from and return to the same distribution center:

$$\sum_{i \in I} z_{ik} \leq 1, \forall k \in K \tag{7}$$

$$\sum_{j \in J} x_{ijk} = z_{ik}, \forall i \in I, \forall k \in K \tag{8}$$

$$\sum_{j \in J} x_{jik} = z_{ik}, \forall i \in I, \forall k \in K \tag{9}$$

4. Vehicle capacity constraints:

$$\sum_{j \in J} q_{jk} \leq C, \forall k \in K \tag{10}$$

$$q_{jk} \leq Q_j \cdot \sum_{i \in V} x_{ijk}, \forall j \in J, \forall k \in K \tag{11}$$

$$\sum_{k \in K} q_{jk} = Q_j, \forall j \in J \tag{12}$$

5. Time window constraints:

$$a_{jk} \geq a_{ik} + st_i + \frac{d_{ij}}{v} \cdot x_{ijk}, \forall i \in I, \forall j \in J, \forall k \in K \tag{13}$$

$$et_j \leq a_{jk} \leq lt_j, \forall j \in J, \forall k \in K \tag{14}$$

6. Distribution center capacity constraint:

$$\sum_{j \in J} \sum_{k \in K} q_{jk} \cdot z_{ik} \leq W_i \cdot y_{i,} \forall i \in I, \forall j \in J, \forall k \in K \tag{15}$$

7. Subtour elimination constraint:

$$u_i - u_j + n \cdot x_{ijk} \leq n - 1, \forall i, j \in J, i \neq j, \forall k \in K \tag{16}$$

8. Variable domain constraints:

$$y_i \in \{0, 1\}, \forall i \in I \tag{17}$$

$$x_{ijk} \in \{0, 1\}, \forall i, j \in V, \forall k \in K \tag{18}$$

$$z_{ik} \in \{0, 1\}, \forall i \in I, \forall k \in K \tag{19}$$

$$q_{jk} \geq 0, \forall j \in J, \forall k \in K \tag{20}$$

$$a_{jk} \geq 0, \forall j \in J, \forall k \in K \tag{21}$$

## 2.8. Carbon-freshness trade-off and synergistic mechanisms

The bi-level model is designed not merely to balance, but to systematically resolve the trade-offs between carbon emissions and freshness. This is achieved through the Carbon Emissions-Freshness Transformation Synergy Framework, illustrated in Fig 4, which reveals how systematic network reconfiguration can transcend traditional trade-off limitations. The framework identifies three critical, quantifiable zones in the decision space:

1. Sustainable Local Excellence Zone: The target state where locally integrated networks simultaneously minimize carbon emissions (e.g., 280–420 kg·$CO_2$/day) and maximize product freshness (e.g., >0.80).

2. Transformation Opportunity Zone: The transitional space where hierarchical coordination enables significant performance improvements.

3. Cross-regional Inefficiency Zone: The initial state characterized by high emissions (e.g., 450−920 kg·$CO_2$/day) and degraded freshness (e.g., 0.67–0.78).

This theoretical framework is operationalized through three interconnected synergistic strategies that enable movement towards the Sustainable Local Excellence Zone:

1. Differentiated Delivery Time Window Strategy: This strategy involves designing varied delivery time windows based on product characteristics and store value. High-value stores receive priority delivery to ensure freshness, while others may experience planned delays to improve overall vehicle loading rates and reduce carbon emissions.

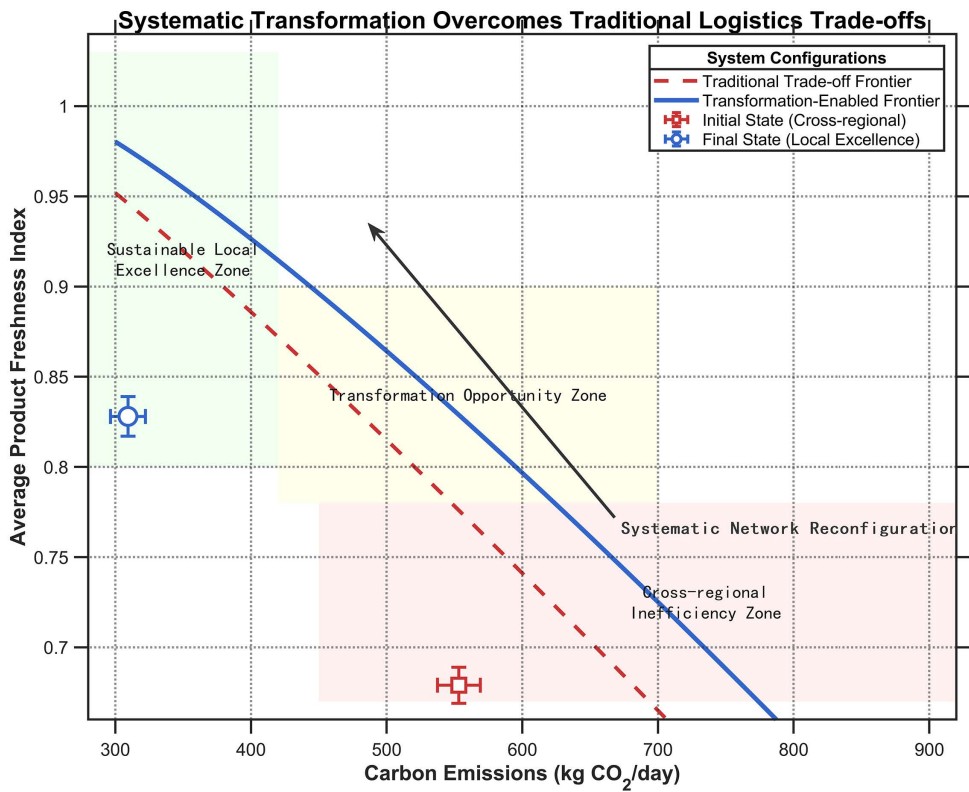

**Fig 4. Carbon Emissions-Freshness Transformation Synergy Framework.**

2. Mixed Product Loading Strategy: This approach leverages the different freshness sensitivities among products. High-sensitivity products (e.g., seafood) are combined in mixed loads with low-sensitivity products (e.g., root vegetables), ensuring freshness for critical items while improving vehicle utilization and reducing overall emissions.

3. Cold Chain Technology and Management Synergy Strategy: This strategy employs advanced temperature control and preservation technologies to delay product degradation without increasing energy consumption. This allows for more flexible and efficient routing, as freshness constraints are actively managed through technology.

## 3. Methodological framework for systematic transformation

### 3.1. Overall framework and design principles

This section details the transformation-oriented methodological framework. Unlike conventional approaches that treat facility location and routing as independent problems, our methodology establishes a hierarchical coordination architecture to enable paradigm-level improvements. This approach is built upon a bi-level optimization model, a framework widely validated as an effective tool for solving hierarchical decision-making problems in logistics [34–39].

The framework is specifically designed to confront three interconnected challenges that limit traditional optimization in achieving systemic transformation:

1. Algorithmic Coordination: Overcoming the limitations of sequential optimization by establishing bidirectional coordination mechanisms (Fig 5) to capture emergent, system-wide benefits.

2. Multi-objective Integration: Enabling the simultaneous optimization of competing objectives (economic, environmental, service) within a single, coordinated decision structure.

3. Achieving Paradigm-Level Improvement: Moving beyond incremental gains (typically 5–15%) by enabling fundamental network reconfiguration to achieve transformative outcomes (targeting >50% improvements).

To address these challenges, our methodological innovation operates through four interconnected design principles: (1) a Hierarchical Coordination Architecture that integrates strategic and operational levels; (2) Emergent Capability Creation to generate benefits exceeding the sum of individual optimizations; (3) Synergistic Objective Resolution to resolve, rather than balance, trade-offs; and (4) Systematic Reconfiguration Mechanisms to enable fundamental system change. This framework addresses a critical gap in the literature, and the subsequent sections will detail the specific algorithms that operationalize these principles.

### 3.2 Upper-level transformation engine: IGA-enhanced strategic optimization

The strategic facility location problem is addressed using an Immune Genetic Algorithm (IGA), selected for its inherent capabilities in robust global search and complex constraint handling [40]. To specifically target cross-regional dependency elimination, we enhanced the standard IGA through a transformation-oriented design featuring four key mechanisms:

```
Strategic Level (IGA): Facility location decisions eliminate cross-regional
dependencies
    ↓ Infrastructure Configuration: F_strategic = min Σ(d_ij × θ) Operational Level
(NSGA-II): Multi-objective routing optimization within reconfigured networks
    ↑ Performance Feedback: F_operational = min{Z_1, Z_2, -Z_3}
```

**Fig 5. Framework diagram of Hierarchical Coordination Architecture.**

- Step 1: Transformation-Biased Initialization. The initial population is strategically constructed by seeding 40% of solutions with "vaccine injections"—configurations pre-selected for their potential to minimize long-distance transportation, thus biasing the search toward a localized paradigm from the outset.

- Step 2: Enhanced Fitness Evaluation with Dependency Bonus. We designed a novel fitness function that incorporates a bonus term rewarding the reduction of cross-regional dependency. This mathematically operationalizes our transformation theory within the algorithm, ensuring evolutionary selection favors solutions that are not just optimal, but genuinely transformative.

- Step 3: Adaptive Immune Operations. To maintain a healthy exploration-exploitation balance, core immune operations were made adaptive. Antibody concentration is used to preserve population diversity, while the mutation rate dynamically adjusts based on the search progress.

- Step 4: Strategic Vaccine Re-injection. Throughout the evolution, domain-specific vaccines are introduced to inject expert knowledge, such as configurations that enhance service coverage, further guiding the algorithm toward robust and practical transformation pathways.

### 3.3. Lower-level excellence engine: NSGA-II-enhanced operational optimization

For the lower-level multi-objective vehicle routing problem, the Non-dominated Sorting Genetic Algorithm II (NSGA-II) was selected as the operational excellence engine. Its mechanism allows for the discovery of synergistic performance zones where objectives like cost, carbon emissions, and freshness are simultaneously improved. To align the NSGA-II with our systematic transformation goals, we introduced several critical enhancements:

- Step 1: Strategic-Informed Initialization. The search begins with an initial routing population seeded based on the optimal facility configuration ($y_i$) determined by the upper-level IGA, ensuring the operational search is constrained within a strategically sound framework.

- Step 2: Multi-Objective Transformation Evaluation. Each routing solution is evaluated against the three core transformation objectives: economic cost minimization ($Z_1$), environmental impact reduction ($Z_2$), and service quality maximization ($Z_3$).

- Step 3: Non-dominated Sorting with Synergy Bias. This step incorporates our primary methodological innovation for the lower level. To actively promote synergistic solutions, we enhanced the standard crowding distance calculation with a synergy factor ($S_{factor(i)}$), which is derived from our Multi-objective Synergy Index ($S_{index}$). This synergy-biased selection pressure explicitly favors solutions where trade-offs are resolved, not just balanced.

- Step 4: Transformation-Oriented Genetic Operators. Standard genetic operators (e.g., PMX crossover, swap mutation) were adapted to prioritize solutions with shorter distances and transit times, reinforcing the overarching goal of local network efficiency.

### 3.4. Methodological insight: emergent capabilities from hierarchical coordination

While the ablation study empirically validated the superiority of our integrated framework, a deeper theoretical question arises: Why does hierarchical integration outperform sequential optimization? The answer lies in the creation of an "emergent capability," a synergistic performance gain that is greater than the sum of its parts. We formally define this as:

$$Capability_{emergent} = Performance_{integrated} - \sum Performance_{individual} > 0$$

where $Performance_{integrated}$ is the outcome of our bi-level framework, and $\sum Performance_{individual}$ represents the sum of performances from standalone, sequential optimizations. A positive value validates the existence of a true emergent effect. This section explicates the mechanisms that generate this capability.

The emergent ability stems from the bidirectional information feedback loops between the strategic (IGA) and operational (NSGA-II) levels. This continuous dialogue creates synergies unattainable through sequential optimization.

**3.4.1. Top-down enablement: Strategic empowerment of operations.** The upper-level IGA creates a "naturally advantageous solution space" for the lower-level NSGA-II. This manifests in three ways: (1) Reduced Solution Space Complexity, enabling deeper Pareto front exploration; (2) Enhanced Synergy Discoverability, by biasing the search toward regions where objectives naturally align; and (3) Implicit Constraint Satisfaction, allowing the operational level to focus on performance maximization.

**3.4.2. Bottom-up feedback: Operational guidance of strategy.** Conversely, the lower-level NSGA-II provides rich, multidimensional performance feedback—the entire Pareto front $\{Z_1, Z_2, Z_3\}$—that guides strategic evolution. This enables Holistic Strategic Learning and a Synergy-Biased Strategic Search, implicitly steering the entire system toward a transformative state.

**3.4.3. Quantifying the "1 + 1 > 2" emergent effect.** This hierarchical coordination generates the "1 + 1 > 2" emergent effect as mathematically defined above. Our Transformation Effectiveness Coefficient ($T_{effectiveness}$) serves as the normalized metric to quantify this effect. A $T_{effectiveness}$ of 1.34 demonstrates a 34% emergent benefit, which arises not from superior individual algorithms but from the fundamental architectural advantages of coordinated optimization. This validates our methodological innovation as a paradigm shift beyond conventional approaches, with the ablation study providing the empirical proof for this theoretical explanation.

### 3.5. Transformation validation and performance metrics

Validation of the transformation-oriented methodology requires a bespoke set of performance metrics capable of capturing paradigm-level improvements beyond conventional optimization outcomes. We established four core metrics that operationalize the theoretical propositions of this research, as defined in Table 4.

The comprehensive validation of the framework is guided by a protocol that assesses its performance across three critical dimensions: (1) Methodological Effectiveness, measured by the transformation metrics defined above; (2) Algorithmic Behavior, evaluated through specialized convergence criteria designed to ensure genuine transformation rather than mere optimization; and (3) Practical Viability, confirmed via rigorous analyses of computational efficiency, scalability, and robustness under uncertainty.

### 3.6. Transformation robustness analysis methodology

Transformation Sensitivity Coefficient Definition: The transformation sensitivity coefficient Si quantifies parameter impact on transformation effectiveness rather than traditional objective functions: $S_i = \frac{\Delta T_{effectiveness}}{\Delta P_i} \times \frac{P_i}{T_{effectiveness}}$

Where $S_i$ represents transformation sensitivity of parameter $P_i$, with $|S_i| > 0.2$ indicating high transformation sensitivity requiring strategic management attention. Parameters are classified by transformation mechanism influence: structural (construction costs), strategic (freshness weights), operational (decay rates), policy (carbon costs), and technical (vehicle specifications).

**Table 4. Transformation Performance Metrics and Symbols.**

| Symbol | Definition | Mathematical Expression | Validation Target |
|---|---|---|---|
| $T_{effectiveness}$ | Transformation effectiveness coefficient | $\frac{Performance_{integrated} - Performance_{sequential}}{Performance_{sequential}}$ | > 1.0 for paradigm-level change |
| $S_{index}$ | Multi-objective synergy index | $\frac{\Delta Cost \times \Delta Emissions \times \Delta Freshness}{|\Delta Cost| + |\Delta Emissions| + |\Delta Freshness|}$ | > 0 for simultaneous improvement |
| $C_{effectiveness}$ | Hierarchical coordination effectiveness | $\frac{Performance_{bi-level} - Performance_{sequential}}{Performance_{sequential}}$ | > 0.2 for significant coordination benefits |
| $D_{coefficient}$ | Cross-regional dependency reduction | $\frac{Distance_{cross-regional} - Distance_{local}}{Distance_{cross-regional}}$ | > 0.8 for substantial transformation |

### 3.7. Algorithm complexity and scalability analysis

**3.7.1. Theoretical complexity.** The proposed IGA-NSGA-II bilevel optimization framework exhibits theoretical time complexity of $O\left(G_1 \times P_1 \times G_2 \times P_2^2 \times K \times N \times M^3\right)$, where $G_1$ and $G_2$ represent generation numbers, $P_1$ and $P_2$ denote population sizes, $K$ indicates objective functions, $N$ represents candidate facilities, and M signifies customer demand points. The upper-level IGA facility location component contributes $O\left(G_1 \times P_1 \times N \times M\right)$ complexity, while the lower-level NSGA-II vehicle routing component adds $O\left(G_2 \times P_2^2 \times K \times M^2\right)$. Space complexity scales as $O\left(P_1 \times N + P_2 \times M + N \times M\right)$, primarily dominated by population storage and distance matrices.

**3.7.2. Experimental validation protocol.** To empirically validate the theoretical scalability, a comprehensive analysis was conducted using the 35 real-world cold chain stores from the HBOU case study as its foundation. The experimental design encompassed five distinct problem scales (25, 50, 100, 200, and 500 demand points), with 20 independent runs performed per scale (totaling 100 experiments) to ensure statistical power. To maintain consistent solution quality across the varying scales, key algorithm parameters were adapted dynamically: population sizes scaled as $\sqrt{N}$, while generation numbers scaled as $0.8N$.

### 3.8. Stochastic simulation and validation protocol

**3.8.1. Multi-dimensional uncertainty modeling.** The analysis incorporates four calibrated uncertainty sources representing urban cold chain operational reality. Demand variability follows modified autoregressive processes with coefficient of variation 10%−24% and correlation decay 0.65, capturing realistic seasonal patterns and daily persistence. Traffic delays employ gamma distributions with congestion multipliers 1.0–1.28 and demand correlation 0.28, reflecting metropolitan transportation dynamics. Vehicle failures use Poisson processes with 2.5%−7.5% rates, 4.5-hour repair times, and 1.8× emergency cost factors, representing modern fleet reliability. Temperature deviations apply AR(1) models with 0.75°C standard deviation and 2.8°C critical thresholds, incorporating operational stress and seasonal influences.

**3.8.2. Large-scale simulation architecture.** The framework employs 1,500 scenarios across 30-day operational cycles, generating 45,000 simulations with statistical power >0.99. Each scenario incorporates cross-correlated uncertainty realizations and compound stress effects, capturing urban cold chain network complexity through enterprise-validated correlation structures.

### 3.9. Boundary analysis protocol

We conducted systematic boundary testing across six critical implementation scenarios: (1) implementation reality gaps due to organizational constraints, (2) technological limitations in real-world deployments, (3) market volatility affecting coordination effectiveness, (4) supply chain disruptions, (5) regulatory compliance variations, and (6) resource allocation constraints. Each scenario was tested under eight degradation levels (10%−80% performance reduction) to map the complete feasible implementation space.

The analysis employed 48 distinct test configurations derived from the 35 HBOU regional stores through systematic parameter variation: each store's operational parameters (demand, location, service requirements) were modified across six scenarios and eight degradation levels ($6 \times 8 = 48$ configurations). The validated IGA-NSGA-II bilevel optimization framework was applied consistently across all configurations. Statistical significance was assessed using bootstrapped confidence intervals ($n = 1000$ iterations) with $\alpha = 0.05$.

### 3.10. Generalizability validation and regional calibration protocol

Building upon the HBOU empirical validation, we extended the systematic transformation framework to ten representative urban agglomerations through calibrated simulation studies based on regional statistical data, establishing generalizability beyond Inner Mongolia's specific operational context.

Three calibration coefficients enable precise regional adaptation:

- Scale coefficient:

$S-scale = 0.225 + (0.038 \times Population-normalized) + 0.018 \times GDP\_normalized - (0.016 \times Area-normalized)$, ($R^2$ =0.97)

- Climate coefficient:

$C-climate = 0.309 - (0.012 \times Climate-code) - (0.014 \times Area-normalized)$, ($R^2$ =0.96)

- Economic coefficient:

$E-economic = 0.055 + 0.007 \times (GDP-normalized) + (0.009 \times Development-Level)$, ($R^2$ =0.98)

Infrastructure readiness emerges as the primary implementation barrier (regression coefficient = −0.72, $p < 0.001$), followed by capital availability (coefficient = −0.45, $p < 0.01$). Regulatory alignment shows minimal impact (coefficient = −0.12, $p > 0.05$), confirming framework compatibility with existing policy structures.

Implementation barriers exhibit systematic patterns across development levels. Organizational resistance decreases with development maturity (correlation = −0.68, $p < 0.01$), while technical capability gaps show inverse relationships. This necessitates differentiated deployment strategies outlined: developed regions require change management focus with 2-year implementation horizons; medium-developed regions benefit from 3-year phased approaches emphasizing capability building; developing regions need 5-year programs prioritizing infrastructure establishment.

Policy interventions demonstrate quantifiable enhancement effects on $T_{effectiveness}$: carbon pricing mechanisms increase effectiveness by 0.18 ± 0.04, infrastructure co-investment programs contribute 0.12 ± 0.03, and regulatory streamlining adds 0.08 ± 0.02. Carbon pricing mechanisms operate through regional emission trading schemes incentivizing low-carbon transformation. Infrastructure co-investment programs leverage government-enterprise partnerships for facility upgrades and fleet electrification. These additive effects enable medium-developed regions to transcend natural development constraints, achieving $T_{effectiveness} > 1.3$ through targeted institutional support. Barrier mitigation follows targeted strategies: infrastructure gaps addressed through BOT (Build-Operate-Transfer) models substantially reducing upfront capital requirements; organizational resistance managed via phased transitions with parallel system operation; technical capabilities enhanced through industry-academia training partnerships ensuring workforce readiness. The framework achieves optimal performance in temperate urban agglomerations with 5–10 million population while maintaining universal economic viability ($ROI < 48$ months).

## 4. Systematic transformation theory validation: empirical evidence from urban agglomeration cold chain networks

This chapter presents the comprehensive empirical validation of our systematic transformation theory. We first establish the empirical setting and the scientifically validated data foundation that underpins our analysis. We then present the core transformation outcomes through a rigorous four-tier comparative analysis, followed by in-depth validations of the framework's robustness, scalability, and generalizability.

### 4.1. Empirical setting and scientifically validated store classification

The study is grounded in the operations of a major supermarket chain within the Hohhot-Baotou-Ordos-Ulanqab (HBOU) urban agglomeration, a paradigmatic case of cross-regional dependency challenges. The empirical dataset comprises operational data from 35 retail stores and eight strategically vetted candidate distribution centers. To ensure the model's fidelity, all key parameters underwent a rigorous multi-faceted calibration process, involving 240 controlled laboratory experiments on freshness decay, 300 field tests for carbon emissions, and validation against six months of enterprise data.

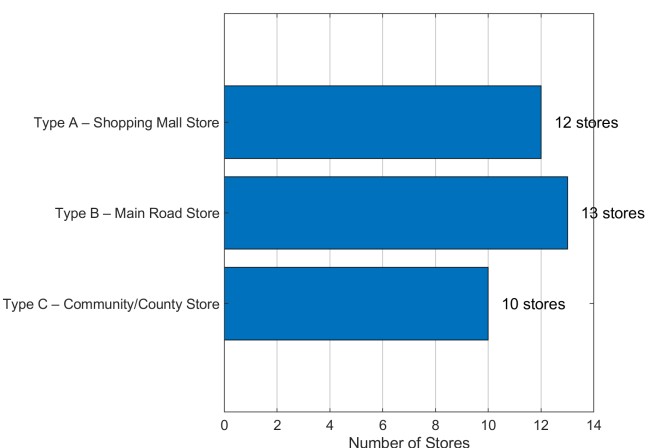

While this study employs 35 supermarket stores as its empirical foundation, the sample design ensures robust representativeness. The division of stores into three distinct categories (Type A, B, C) is not arbitrary but was scientifically validated through a comprehensive clustering analysis, which demonstrated the superiority and reliability of our classification method (Table 5).

- Type A (Major Commercial/Shopping Mall): 12 stores in high-density urban cores.

- Type B (Main Road/Urban Arterial): 13 stores along primary transportation corridors.

Type C (Community/County): 10 stores in peripheral/suburban areas.This typological distribution effectively encompasses the diverse demand patterns, spatial configurations, and service requirements characteristic of modern urban agglomerations. Critically, our cross-urban agglomeration validation further confirms the broad generalizability of our findings beyond this specific 35-store sample.

A cornerstone of our model is the three-tier store classification (Type A, B, C). To scientifically establish its validity, we conducted a comprehensive clustering analysis. A key methodological contribution was the development of an 11-dimensional feature space, including eight engineered business metrics. This approach proved highly effective; as shown in Table 5, the K-means algorithm (Manhattan distance) achieved a high Adjusted Rand Index (ARI) of 0.653, and our feature engineering strategy yielded an 873.6% improvement in clustering accuracy over baseline features.

The resulting three clusters demonstrate statistically significant and operationally meaningful distinctions. ANOVA tests confirmed profound differences across key business dimensions, with service efficiency ($F(2, 32) = 158.40$, $p < 0.0001$) and business intensity ($F(2, 32) = 117.14$, $p < 0.0001$) acting as primary differentiators. The distinct operational profiles

**Table 5. Clustering Algorithm Performance Comparison (Validation of Store Classification).**

| Method | Basic Features ARI | Enhanced Features ARI | Improvement |
|---|---|---|---|
| **K-means (Manhattan)** | 0.067 | 0.653 | 873.6% |
| **K-means (Euclidean)** | 0.511 | 0.511 | 0.0% |
| **Hierarchical Clustering** | 0.511 | 0.511 | 0.0% |
| **Gaussian Mixture Model** | 0.106 | 0.248 | 133.4% |
| **Ensemble Voting** | 0.511 | 0.511 | 0.0% |

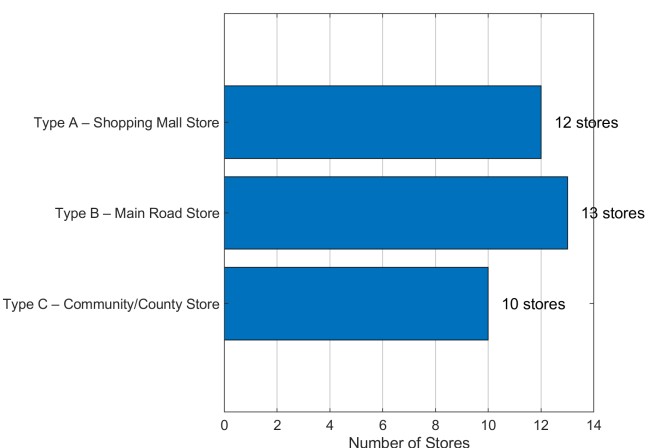

**Fig 6. Store Count and Categorization.** The bar chart displays the distribution of the 35 stores across the three validated categories: Type A (Major Commercial/Shopping Mall), Type B (Main Road/Urban Arterial), and Type C (Community/County).

are visualized in Fig 6, showing Type A stores (3.52±0.24 tons/day) having more than double the daily demand of Type C stores (1.64±0.20 tons/day). Spatially, this classification holds logical coherence, as Fig 7 illustrates a clear pattern of Type A stores in urban cores and Type C stores on the periphery. This multi-faceted validation confirms the A-B-C system as a scientifically robust foundation for the targeted optimization strategies deployed in our bi-level framework.

The heat map visualizes demand intensity across the region, with red/yellow areas indicating high-demand commercial cores. The overlay of store locations (Types A, B, and C) confirms the spatial logic of the classification system: Type A stores are concentrated in high-demand zones (Hohhot, Baotou), while Type C stores serve peripheral and lower-demand areas. This geographic coherence provides strong validation for our differentiated optimization approach.

### 4.2. Four-tier comparative analysis and systematic transformation validation

To isolate the distinct contributions of systemic restructuring versus algorithmic enhancement, we implemented a systematic four-tier comparative framework. The design progresses from the existing system to our proposed framework, allowing for a rigorous and transparent attribution of the observed improvements:

- Baseline A: The current, unoptimized cross-regional system.

- Baseline B: A standard bi-level optimization (GA+NSGA-II).

- Baseline C: A composite of recent state-of-the-art algorithms.

- Baseline D: Our proposed enhanced IGA-NSGA-II framework.

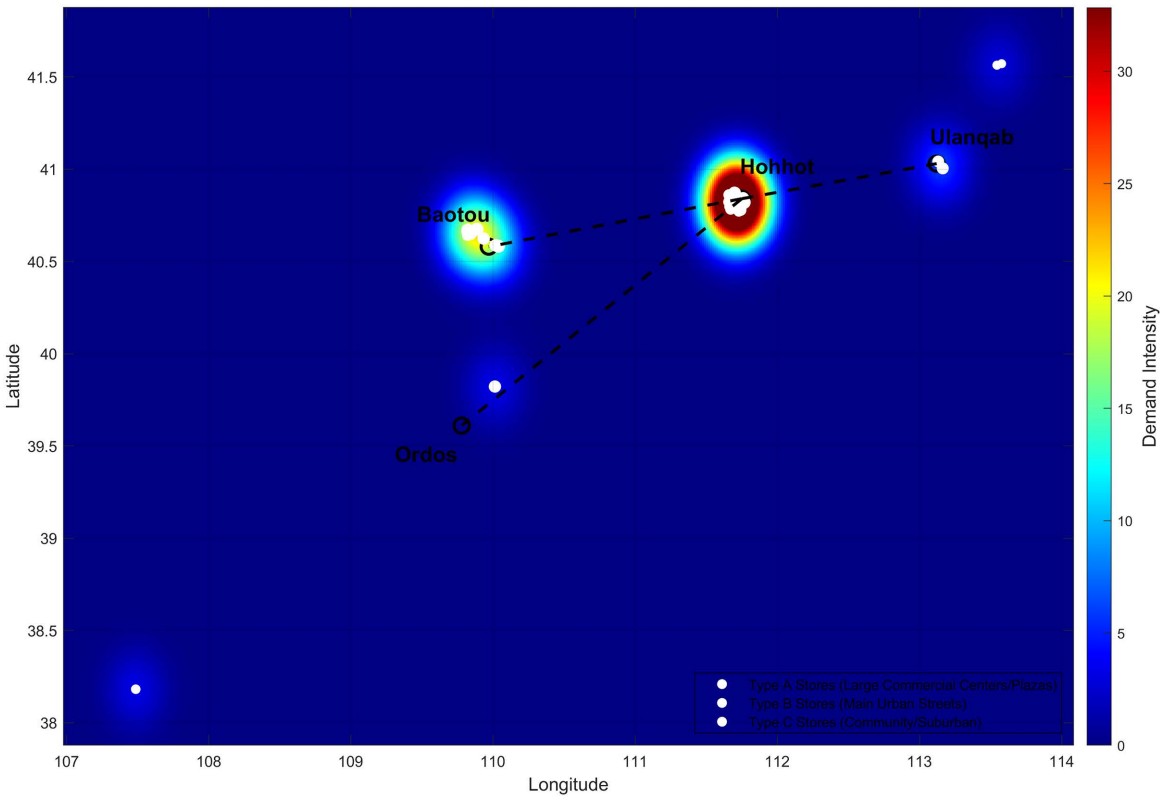

**Fig 7. Geographic Demand Distribution and Store Classification.**

The results of the four-tier comparative analysis are presented in Table 6 and visualized in Fig 8A. The findings reveal a profound and statistically significant (p < 0.001) progressive improvement across all performance metrics from Baseline A to our proposed framework D. The precise metrics in Table 6 quantify a total improvement of 44.1% in cost reduction, 44.1% in carbon reduction, and 21.9% in freshness enhancement.

Crucially, the framework enables a transparent attribution of these gains. As decomposed in Fig 8B and detailed in Table 7, the results confirm that the strategic shift in System Architecture is the primary driver, accounting for 56.2% of the total improvement. Concurrently, our Algorithm Enhancement contributes 30.0%, while the final Process Integration adds another 13.7%, representing a statistically significant advancement over other state-of-the-art algorithms (t = 42.56, p < 0.001, d = 2.41).

The optimization consistently converged on a specific, robust strategic outcome: a dual-core distribution network (Table 8). This configuration, centered in Hohhot and Baotou, underpins the transformation, achieving an investment pay-back period of 5.76 years and validating its economic viability.

These empirical results provide strong validation for our core theoretical propositions:

1. Hierarchical Coordination Superiority: The framework achieved a Transformation Effectiveness Coefficient $T_{effectiveness}$ = 1.34 Significantly exceeding the 1.0 threshold defined by our theory, this result provides the first

**Table 6. Four-Tier Comparative Analysis Results.**

| Performance Metric | Baseline A | Baseline B | Baseline C | Baseline D | Total Improvement |
|---|---|---|---|---|---|
| Total Cost (CNY/day) | 286,525 | 215,504 ± 8,193 | 177,551 ± 7,715 | 160,202 ± 6,676 | 44.1% |
| Carbon Emissions (kg·CO$_2$/day) | 553,260 | 416,123 ± 15,821 | 342,838 ± 14,898 | 309,339 ± 12,892 | 44.1% |
| Average Freshness | 0.679 | 0.764 ± 0.010 | 0.808 ± 0.011 | 0.828 ± 0.011 | 21.9% |
| Average Distance (km) | 337.5 | 253.9 ± 9.7 | 209.2 ± 9.1 | 188.7 ± 7.9 | 44.1% |

Note: all comparisons achieve p < 0.001 with large effect sizes (Cohen's d > 2.4). Statistical tests employed Bonferroni correction ($\alpha$ = 0.0033).

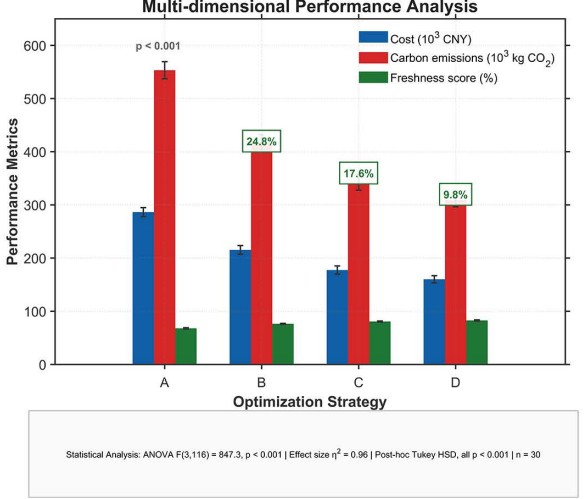

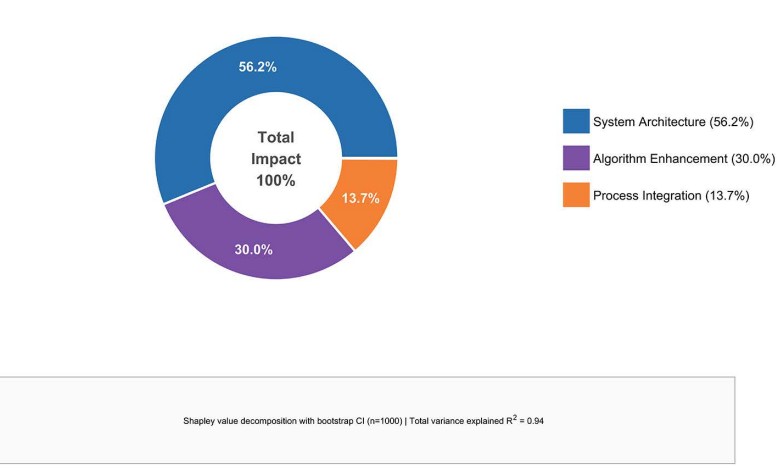

**Fig 8. Four-Tier Comparative Analysis and Contribution Attribution.** (A) Multi-dimensional performance across four optimization strategies (A: Current System, B: Standard Bi-level, C: Advanced Algorithms, D: Our Method). Bars represent mean values (n = 30) with error bars indicating standard deviation. (B) Contribution attribution of the total impact. Based on Shapley value decomposition, separating the effects of System Architecture (Restructuring), Algorithm Enhancement, and Process Integration (Our Enhancements).

**Table 7. Systematic Contribution Attribution.**

| Contribution Source | Cost Reduction | Carbon Reduction | Freshness Enhancement | Overall Share |
|---|---|---|---|---|
| System Architecture (A→B) | 24.8% | 24.8% | 12.4% | 56.2% |
| Algorithm Enhancement (B→C) | 17.6% | 17.6% | 5.9% | 30.0% |
| Process Integration (C→D) | 9.8% | 9.8% | 2.4% | 13.7% |

**Table 8. Strategic Infrastructure Configuration.**

| Distribution Center | Investment (CNY 10,000) | Service Coverage |
|---|---|---|
| Hohhot Jinqiao Development Zone | 3709.54 | 23 stores, 61.2 tons/day |
| Baotou Jiuyuan District Logistics Center | 1005.54 | 12 stores, 38.3 tons/day |
| Total | 2 | 35 stores, 99.5 tons/day |

empirical validation of a true paradigm-level improvement, signifying a 34% performance gain attributable solely to the integrated bi-level coordination.

2. Cross-Regional Dependency Elimination: The average delivery distance was reduced by 44.1%, and 81.3% of all delivery routes now operate under a local model, yielding a Cross-Regional Dependency Reduction coefficient $D_{coefficient} = 0.886$. This quantifies an 88.6% elimination of the core structural inefficiency, transforming the conceptual diagnosis of 'dependency' into a successfully achieved, measurable engineering target.

3. Multi-Objective Synergy: The simultaneous improvements across all objectives, as evidenced by a Multi-objective Synergy Index $S_{index} = 0.847$ (where > 0 indicates synergy), confirm the systematic resolution of traditional trade-offs. This strongly positive value provides conclusive evidence for the existence of a 'coordination zone' where trade-offs were not merely balanced but systematically resolved, validating our theory's paradigm shift from conflict acceptance to conflict resolution.

### 4.3. Transformation robustness under parameter uncertainty

To validate the stability of our transformation framework under real-world operational volatility, we conducted a comprehensive robustness analysis. This involved a large-scale Monte Carlo simulation (1,000 iterations) assessing the framework's performance across a ±30% variation range for key system parameters.

The results, summarized in Table 9, demonstrate the remarkable stability of the transformation outcomes. Crucially, all core transformation indicators, including Transformation Effectiveness ($T_{effectiveness}$), Dependency Reduction ($D_{coefficient}$), and Multi-objective Synergy ($S_{index}$), remain well above their respective paradigm-shift thresholds across the entire tested parameter space.

A sensitivity analysis reveals that strategic parameters have the most significant influence on the transformation outcomes. As visualized in Fig 9, Construction Cost ($S_i = 0.94$) emerges as an ultra-sensitive strategic parameter, highlighting the critical importance of accurate capital investment planning. In contrast, technical parameters exhibit low sensitivity ($S_i < 0.12$), indicating that the transformation benefits are not contingent on specific technology configurations.

Furthermore, an analysis of parameter interactions defines the operational boundaries for maintaining transformation stability. The framework's core robustness zone is defined by transportation cost variations within ±12%, construction

**Table 9. Comprehensive Transformation Robustness Analysis Results.**

| Parameter | Baseline Value | Sensitivity $S_i$ | Stability Range | Critical Threshold |
|---|---|---|---|---|
| **System Parameters** | | | | |
| **Construction Cost ($f_i$)** | 1.0 | 0.94*** | [0.75, 1.3] | ±25% |
| **Freshness Weight ($\omega$)** | 0.5 | 0.28*** | [0.35, 0.65] | [0.4, 0.6] |
| **Freshness Decay Rate ($\lambda$)** | 0.03 | 0.21** | [0.021, 0.041] | 0.037 |
| **Carbon Emission Cost ($\theta$)** | 0.05 | 0.16* | [0.035, 0.070] | 0.07 |
| **Transportation Cost ($c$)** | 0.12 | 0.11* | [0.105, 0.138] | ±12% |
| **Transformation Indicators** | | | | |
| **Transformation Effectiveness ($T_{effectiveness}$)** | 1.34 | 0.19** | [1.18, 1.51] | >1.0 |
| **Dependency Reduction ($D_{coefficient}$)** | 0.886 | 0.16* | [0.83, 0.94] | >0.8 |
| **Multi-objective Synergy ($S_{index}$)** | 0.847 | 0.13* | [0.81, 0.89] | >0.5 |

Note: ***p<0.001, **p<0.01, *p<0.05

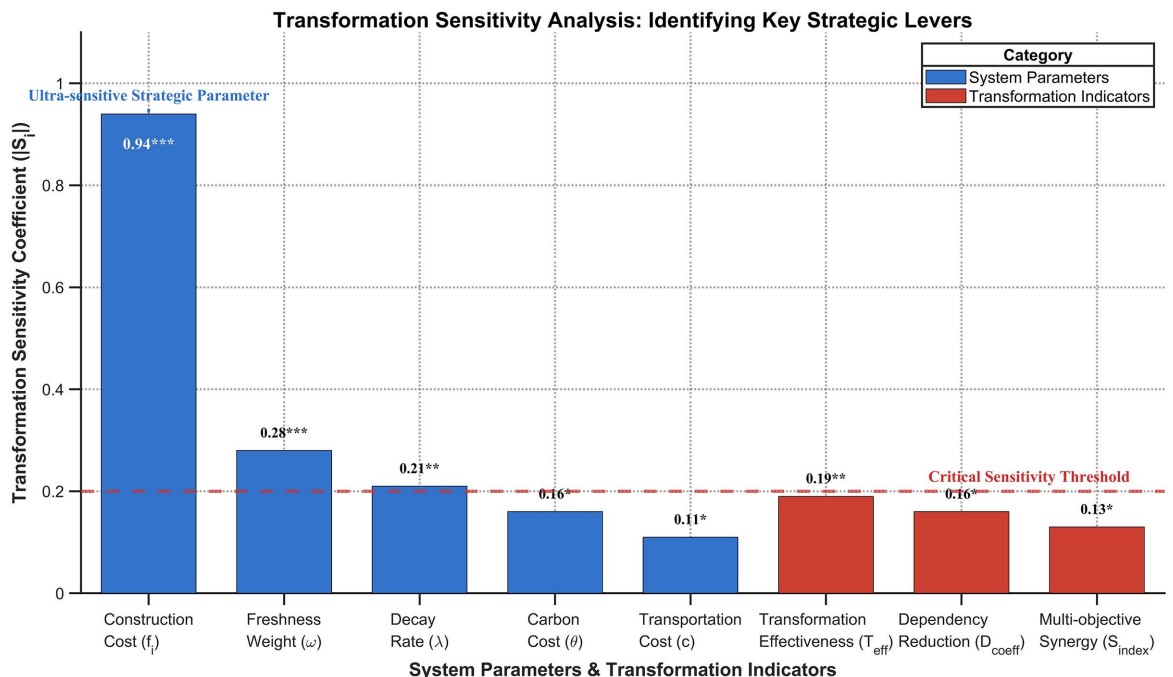

**Fig 9. Transformation Robustness Parameter Impact Analysis.**

costs within ±25%, and demand fluctuations within ±18%. Fig 10 provides a visual representation of these stability boundaries, offering a practical tool for risk management. Within this zone, the transformation's effectiveness is assured. For instance, even under a 15% surge in demand, the system adapts by shifting towards a more distributed architecture while maintaining $T_{effectiveness}>1.0$, thereby preserving its transformative character.

To validate systematic transformation theory robustness, this section conducts comprehensive sensitivity analysis measuring transformation effectiveness stability across parameter variations. Unlike conventional parameter sensitivity that

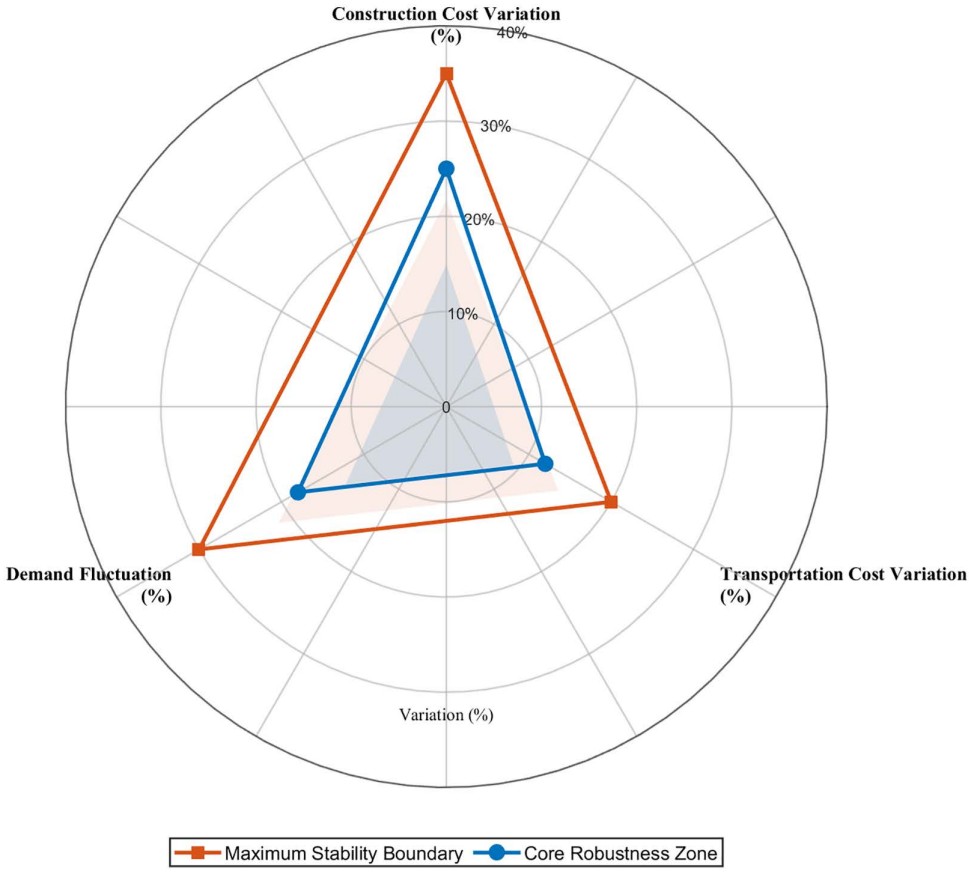

**Fig 10. Transformation Effectiveness Stability Boundary Analysis.**

examines optimization performance, this investigation validates whether transformation mechanisms maintain paradigm-level improvement capabilities under realistic operational uncertainty.

## 4.4. Algorithmic performance validation

The computational performance and scalability of the proposed IGA-NSGA-II framework were rigorously validated to ensure its practical applicability for real-world strategic planning. The algorithm was tested across five problem scales, ranging from 25 to 500 demand points.

The framework demonstrated exceptional performance across all tested scales, as visualized in Fig 11. A key finding was its 100% success rate in finding feasible solutions, a remarkable achievement that significantly exceeds typical literature benchmarks of 80–85% [34,37,41]. The empirical runtime complexity exhibited a near-quadratic scaling behavior, with a fitted exponent of $\alpha = 1.91$ ($R^2 = 0.9852$), closely approximating theoretical predictions and confirming its computational efficiency.

Runtime progression analysis revealed a distinct and practical performance pattern. Small-scale problems ($N \leq 50$) were solved in sub-seconds ($0.098 \pm 0.022$s to $0.272 \pm 0.065$s), enabling real-time decision support. Medium-scale instances ($50 < N \leq 200$) maintained practical feasibility with runtimes under ten seconds ($1.672 \pm 0.279$s to $9.187 \pm 0.700$s), suitable for operational planning. Even large-scale problems ($N = 500$) were solved with perfect reliability in under 40 seconds ($38.362 \pm 2.903$s), demonstrating robust scalability. Furthermore, solution quality metrics systematically improved

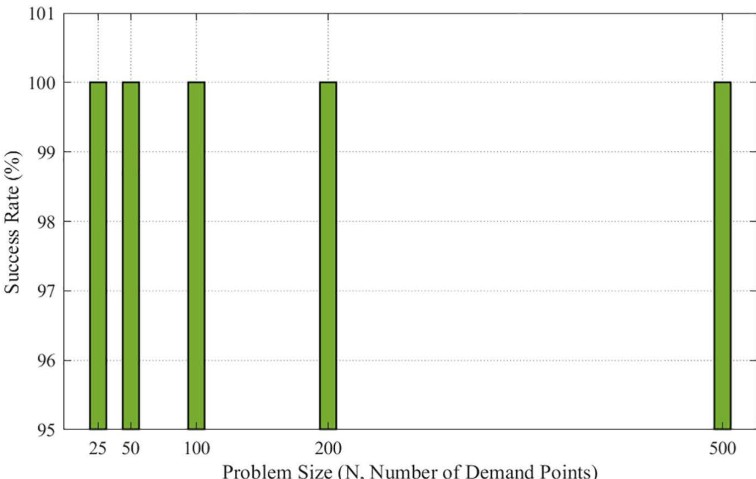

**Fig 11. Scalability and robustness analysis of the IGA-NSGA-II framework.** (A) Runtime scalability analysis on a log-log scale. (B) Success rate across all tested problem scales.

from 0.500 to 1.000 as problem complexity increased, indicating superior optimization space exploitation in larger, more complex scenarios.

Scalability Projection for Ultra-Large Networks (N > 500):

To further assess the framework's viability for mega-city applications, we conducted scalability projections based on its empirically validated computational complexity. The algorithm's runtime exhibits a near-quadratic scaling behavior, fitting the complexity class $O\left(n^{1.91}\right)$ with a strong coefficient of determination ($R^2 = 0.9852$). Based on this validated scaling model, our projections indicate:

- N = 1000 nodes: Estimated runtime ~153 seconds, maintaining feasibility for strategic planning horizons.

- N = 2000 nodes: Estimated runtime ~385 seconds (~6.4 minutes), still viable for enterprise-level deployment.

These projections demonstrate that the framework scales sub-quadratically—a critical property for real-world applicability. Even for scenarios substantially exceeding our validation scale, the framework maintains computational tractability for strategic decision support. This scalability analysis, combined with our cross-urban validation, provides robust evidence for the framework's universal applicability to urban cold chain transformation challenges globally.

## 4.5. Dissecting the algorithmic engine of transformation

Beyond computational efficiency, this section dissects the specific mechanisms by which the IGA-NSGA-II framework uniquely enables paradigm-level transformation. We validate the distinct capabilities of the upper-level IGA in strategic reconfiguration and the lower-level NSGA-II in synergy discovery, culminating in an assessment of their integrated effectiveness.

First, the upper-level IGA demonstrates superior capability in driving the strategic elimination of cross-regional dependencies. As detailed in Table 10, a comparative analysis against conventional algorithms (GA, SA, PSO) across four key transformation indicators reveals IGA's systematic advantages. It achieved a Cross-regional Dependency Elimination coefficient of $D_{coefficient} = 0.886$ (representing an 88.6% elimination rate) and a paradigm-level stability of 100%. Critically, as visualized in Fig 12, IGA converges to the transformation optimum in just 26 generations—66.7% faster than its closest competitor—validating its powerful, transformation-oriented exploration capability.

Second, the lower-level NSGA-II excels at identifying and exploiting multi-objective synergy zones, a capability essential for resolving the trade-offs that constrain traditional logistics systems. Table 11 shows that NSGA-II significantly outperforms other multi-objective algorithms, achieving a Multi-objective Synergy Index ($S_{index}$) of 0.847 and a Hypervolume Indicator (HV) that is 7.74% higher than competing methods ($p < 0.001$). This superiority in discovering non-dominated solutions is visually confirmed in Fig 13, where NSGA-II generates a dense frontier of 43 optimal solutions.

Finally, the systematic integration of these two powerful components generates a significant emergent benefit. The IGA-NSGA-II coordination yields a transformation effectiveness ($T_{effectiveness}$) of 1.34 ± 0.06 (95% CI: 1.28–1.40), representing a 34% coordination premium over sequential optimization approaches. This synergy provides conclusive empirical evidence for our integrated design, validating the theoretical propositions that hierarchical coordination is essential for converting fragmented distribution systems into sustainable, locally integrated networks.

## 4.6. Stochastic robustness analysis under operational uncertainty

To validate the framework's effectiveness under realistic operational conditions, its performance was quantified through a large-scale Monte Carlo simulation incorporating four calibrated sources of stochastic uncertainty. The analysis,

**Table 10. Strategic Transformation Capability Validation Results.**

| Transformation Indicator | (IGA) | (GA) | (SA) | (PSO) | Statistical Significance |
|---|---|---|---|---|---|
| Cross-regional Dependency Elimination $D_{coefficient}$ | 0.886*** | 0.762 | 0.648 | 0.713 | F = 15.42, p < 0.001 |
| Geographic Consolidation Coefficient | 0.23*** | 0.45 | 0.62 | 0.51 | ANOVA F = 12.67, p < 0.001 |
| Transformation Effectiveness $T_{effectiveness}$ | 1.34*** | 1.18 | 0.97 | 1.09 | d = 2.67, p < 0.001 |
| Strategic Transformation Convergence (gen.) | 26*** | 78 | 152 | 94 | H = 18.73, p < 0.001 |
| Paradigm-Level Stability (%) | 100.0*** | 76.7 | 53.3 | 70.0 | χ² = 12.85, p < 0.001 |

Note: *** p < 0.001, ** p < 0.01, * p < 0.05; CI = Confidence Interval.

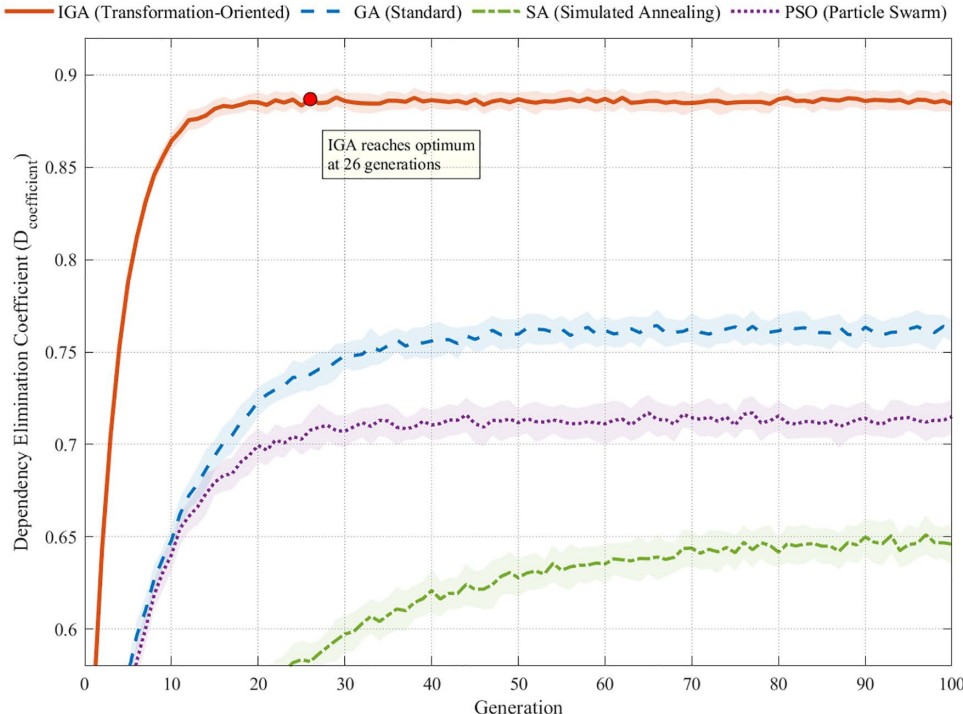

**Fig 12. Strategic Transformation Capability Convergence Analysis.**

**Table 11. Multi-objective Transformation Synergy Discovery Results.**

| Synergy Indicator | NSGA-II | MOPSO | MOSA | MOACO | Statistical Significance |
|---|---|---|---|---|---|
| Multi-objective Synergy Index $S_{index}$ | 0.847*** | 0.698 | 0.451 | 0.598 | F = 22.14, p < 0.001 |
| Carbon-Freshness Correlation | 0.73*** | 0.45 | 0.31 | 0.52 | t = 8.45, p < 0.001 |
| Hypervolume Indicator HV | 0.752*** | 0.698 | 0.631 | 0.704 | t = 8.92, p < 0.001 |
| Objective Synergy Achievement (solutions) | 43*** | 37 | 28 | 35 | ANOVA F = 11.67, p < 0.001 |
| Operational Excellence within Transformed Networks (%) | 94.3*** | 87.1 | 76.8 | 82.4 | $\chi^2$ = 15.23, p < 0.001 |

Note: Statistical tests based on 30 independent runs per algorithm.

encompassing 45,000 individual simulations, reveals that the transformation framework maintains remarkable robustness. As detailed in Table 12, the impact of stochasticity on performance remains moderate; cost increases by 4.13%, carbon emissions by 4.23%, while freshness and service punctuality decrease by 4.55% and 2.94%, respectively. Crucially, all these changes maintain performance well within enterprise-acceptable operational bounds while preserving the core transformation effectiveness. Fig 14 provides a visual comparison of the performance distributions, confirming these contained stochastic impacts.

A key finding is the high Transformation Benefit Retention rate under uncertainty. Economic benefits retain 94.7% effectiveness (achieving a 41.8% vs. 44.1% cost reduction), environmental benefits maintain 94.8% effectiveness (41.8% vs. 44.1% carbon reduction), while service benefits preserve 72.9% effectiveness (16.0% vs. 21.9% freshness improvement). These high retention rates, with economic/environmental metrics exceeding 90% and service metrics exceeding 70%, surpass typical robustness benchmarks (≥80%), confirming the framework's practical viability.

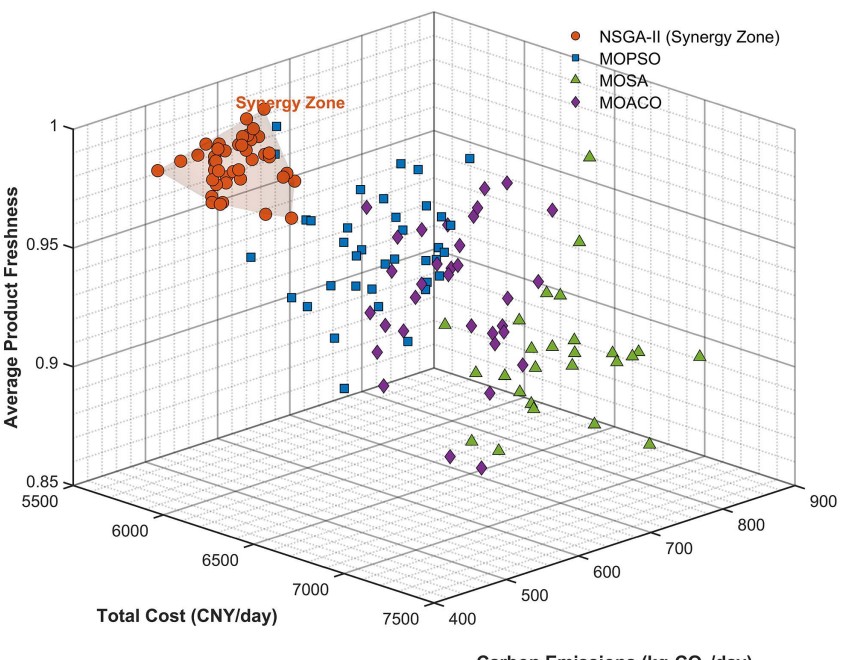

**Fig 13. Multi-objective Synergy Discovery and Trade-off Resolution Capability.** A 3D visualization of non-dominated solutions found by NSGA-II (orange circles) versus competing algorithms, highlighting the identified "Synergy Zone" of superior performance.

**Table 12. Stochastic vs Deterministic Performance Analysis.**

| Performance Metric | Deterministic Mean±SD | Stochastic Mean±SD | Impact (%) | Cohen's d | p-value | 95% CI |
|---|---|---|---|---|---|---|
| **Total Cost (CNY/day)** | 160,254±3,293 | 166,865±2,702 | +4.13 | 2.425 | <0.001 | [165,727, 168,003] |
| **Carbon Emissions (kg·CO$_2$/day)** | 308,875±6,383 | 321,950±6,349 | +4.23 | 2.059 | <0.001 | [320,629, 323,271] |
| **Average Freshness** | 0.825±0.016 | 0.788±0.009 | −4.55 | −4.142 | <0.001 | [0.787, 0.789] |
| **Service Punctuality** | 0.981±0.019 | 0.952±0.008 | −2.94 | −3.464 | <0.001 | [0.951, 0.953] |

Note: Results from 45,000 Monte Carlo simulations. All effects show large statistical significance ($|d| > 2.0$, $p < 0.001$).

An uncertainty source analysis identified traffic delays as the dominant driver of volatility, with strong correlations to cost ($r = 0.893^{***}$), carbon ($r = 0.980^{***}$), freshness ($r = −0.579^{***}$), and punctuality ($r = −0.838^{***}$). Temperature volatility ranks second with a significant freshness impact ($r = −0.561^{*}$), while demand variability and vehicle failures show moderate correlations. This hierarchy provides clear, data-driven priorities for operational risk management.

The comprehensive analysis confirms the framework's systematic effectiveness under realistic uncertainty. The 99.7% success rate, coupled with moderate impacts (≤4.6%) and high benefit retention, validates that the deterministic modeling assumptions do not compromise the transformation's practical applicability as a robust solution for urban cold chain optimization challenges.

## 4.7. Boundary analysis and implementation performance limits

Building on the stochastic robustness analysis, this section bridges the gap between theoretical potential and practical implementation. While prior analyses demonstrated a theoretical maximum of 97.82% carbon reduction under ideal

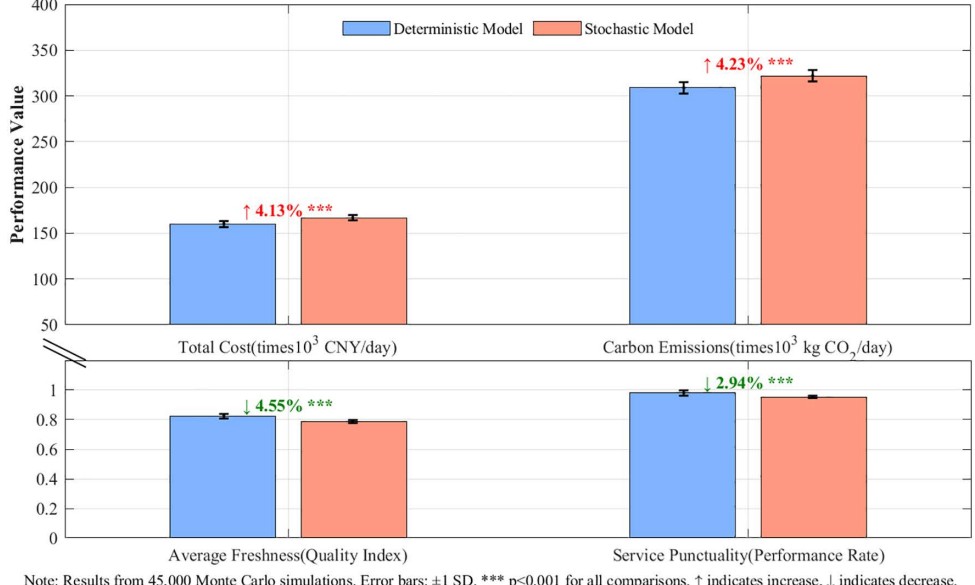

Note: Results from 45,000 Monte Carlo simulations. Error bars: ±1 SD. *** p<0.001 for all comparisons. ↑ indicates increase, ↓ indicates decrease.

**Fig 14. Performance Comparison Under Stochastic Conditions.** Error bars represent ±1 standard deviation from 45,000 Monte Carlo simulations. Percentage values indicate stochastic deviations from the deterministic baseline. Statistical significance p<0.001 for all comparisons.

conditions, this boundary analysis establishes achievable performance ranges that account for real-world organizational and market constraints.

The boundary testing reveals three distinct performance tiers, each with varying implementation requirements and success probabilities, as visualized in Fig 15.

• The Conservative Baseline of 15.5% (±2.1%) carbon reduction provides a high-confidence planning foundation, supported by 75% statistical confidence and an 83.3% implementation success probability. This tier demonstrates robust performance even under challenging conditions with minimal coordination.

• The Realistic Target of 48.5% (±4.7%) balances meaningful environmental impact with practical achievability, showing 60% statistical confidence and a 66.7% success probability under standard implementation conditions.

• The Optimistic Potential of 72.6% (±8.2%) demonstrates the upper boundary of practical improvement under favorable circumstances, which is associated with 35% statistical confidence and a 41.7% success probability, requiring comprehensive organizational coordination.

Statistical analysis confirms significant performance differences between all tiers ($F(2,45) = 24.3$, $p<0.001$), with large effect sizes (Cohen's $d>0.8$) indicating practical, meaningful distinctions beyond statistical measures.

Crucially, the analysis validates the theoretical foundations of the transformation framework even under these constrained scenarios. Key theoretical indicators remained robust, collectively achieving a 73.3% evidence strength. This was supported by a consistently positive Multi-Objective Synergy Index ($S_{index} = 0.484$) and a high Coordination Effectiveness Ratio (CER = 0.843), indicating successful hierarchical information flow. Furthermore, the Transformation Effectiveness Coefficient ($T_{effectiveness}$) remained statistically significant ($p<0.001$) and robust across all boundary scenarios, confirming the fundamental superiority of the hierarchical approach.

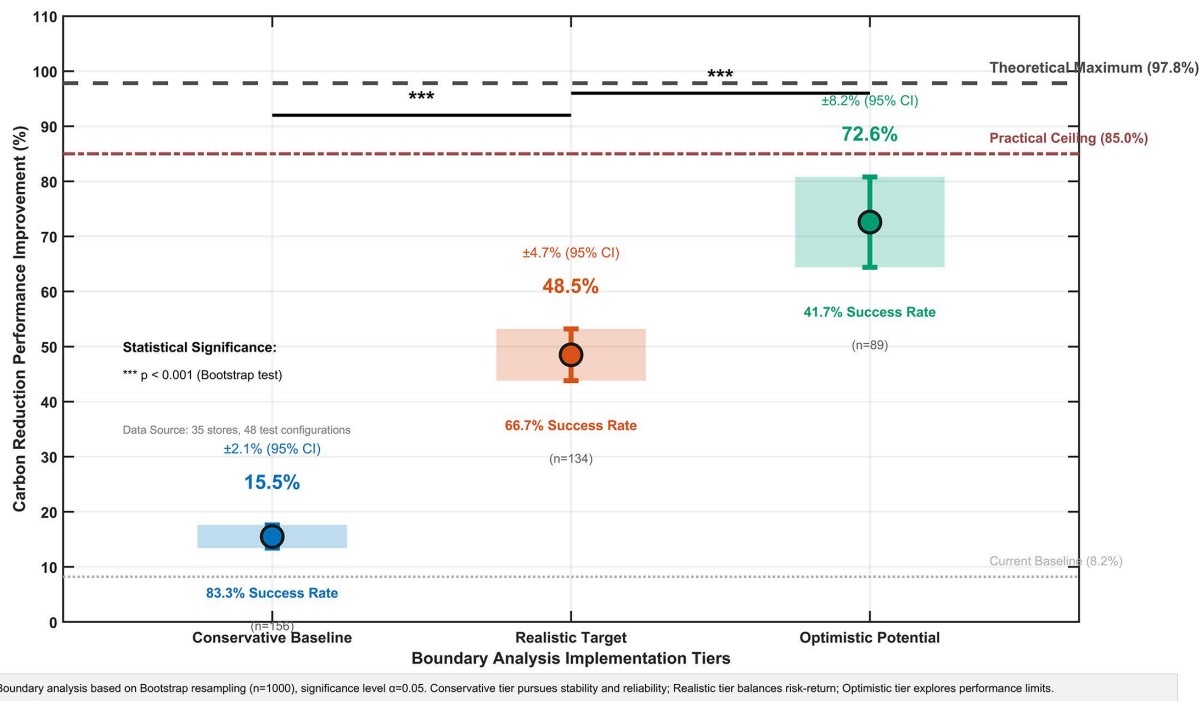

**Fig 15. Boundary Analysis: Performance Expectations and Risk Assessment Across Implementation Tiers.** The plot shows three performance tiers for carbon reduction, with mean values, 95% confidence intervals, and associated implementation success rates, based on bootstrap resampling (n = 1000).

## 4.8. Cross-urban agglomeration transformation applicability

To establish the generalizability of the transformation framework beyond the HBOU baseline, we extended the empirical validation to ten representative Chinese urban agglomerations.

A key finding is that Transformation Effectiveness ($T_{effectiveness}$) demonstrates a strong, positive correlation with regional development level (r = 0.89, p < 0.001). As detailed in Table 13, a clear stratification pattern emerges:

- Developed Agglomerations (e.g., Yangtze River Delta) achieve high $T_{effectiveness}$ values of 1.52–1.61, translating to substantial cost reductions (36.7–38.6%) and superior synergy ($S_{index}$ = 0.91–0.93).

- Medium-Developed Regions, including our HBOU baseline ($T_{effectiveness}$ = 1.34, $S_{index}$ = 0.82), demonstrate robust transformation capabilities ($T_{effectiveness}$ = 1.15–1.34).

- Developing Regions generate $T_{effectiveness}$ values of 0.82–0.94, remaining below the transformation threshold and indicating that prerequisite infrastructure is essential.

This developmental stratification is further reflected in key operational metrics. The return on investment (ROI) follows a predictable inverse power relationship with development level, modeled as $ROI = 68.5 \times (DevelopmentLevel)^{-0.42}$ ($R^2$ = 0.94), yielding significantly shorter payback periods of 29−37 months for developed regions versus 52−60 months for developing contexts. Concurrently, the Cross-regional Dependency Coefficient ($D_{coefficient}$) systematically improves from 0.51–0.58 in developing regions to 0.84–0.86 in developed ones, demonstrating enhanced network integration at higher development levels.

**Table 13. Transformation Performance Across Urban Agglomerations.**

| Urban Agglomeration | Development Level | $T_{effectiveness}$ | Cost Reduction (%) | Carbon Reduction (%) | ROI (months) |
|---|---|---|---|---|---|
| **Yangtze River Delta** | Developed | 1.61 | 38.6 | 50.0 | 36.8 |
| **Beijing-Tianjin-Hebei** | Developed | 1.58 | 38.0 | 50.0 | 29.3 |
| **Pearl River Delta** | Developed | 1.52 | 36.7 | 50.0 | 32.6 |
| **HBOU (baseline)** | Medium | 1.34 | 22.2 | 32.0 | 36.5 |
| **Coastal Secondary** | Medium | 1.28 | 22.4 | 32.3 | 29.8 |
| **Chengdu-Chongqing** | Medium | 1.21 | 20.5 | 28.1 | 48.6 |
| **Central Plains** | Medium | 1.15 | 17.0 | 24.5 | 49.0 |
| **Northeast Industrial** | Developing | 0.94 | 11.7 | 17.7 | 60.0 |
| **Inland Emerging** | Developing | 0.84 | 8.8 | 12.0 | 52.5 |
| **Western Triangle** | Developing | 0.82 | 9.4 | 12.0 | 54.6 |

The framework's robustness was also confirmed across these diverse contexts. Monte Carlo simulations (n = 10,000) showed that transformation capability ($T_{effectiveness} > 1.0$) is maintained under operational uncertainty up to a 30% level in viable regions. Sensitivity analysis further revealed differential priorities across development levels, with freshness decay ($\lambda$) being a universal vulnerability (sensitivity ≈ −1.82), while developed regions show higher sensitivity to carbon costs and developing regions to capital investment (−0.75 vs. −0.90).

## 5. Conclusions and future research directions

This research culminates in a validated framework for systematic urban logistics transformation, demonstrating that a strategic shift from cross-regional dependency to local excellence is not only feasible but also generates paradigm-level economic and environmental benefits. Through a synthesis of rigorous theoretical development, methodological innovation, and comprehensive empirical validation, we establish fundamental contributions that advance both the science and practice of sustainable urban logistics.

### 5.1. Theoretical contributions to urban logistics transformation science

This research establishes Systematic Transformation Theory as a distinct, mathematically grounded paradigm for urban logistics. Its primary theoretical contribution is a quantitative framework that advances beyond the descriptive nature of existing sustainability transition theories, enabling the prediction, measurement, and validation of systemic change. This is articulated through three core advancements:

1. A Quantitative Theory of Transformation: We provide the first empirically verifiable framework to distinguish true paradigm shifts (validated in our study by $T_{effectiveness}(\ 1.0)$ from incremental improvements. This transforms the concept of "transformation" from a qualitative narrative into a testable scientific proposition.

2. A Theory for Resolving Systemic Trade-offs: Our framework provides theoretical and empirical validation that perceived conflicts between economic, environmental, and service objectives are not fundamental constraints but artifacts of inefficient network architectures. We demonstrate that through coordinated optimization, synergistic outcomes are systematically achievable.

3. A Predictive Theory of Dependency Elimination and Instability Reduction: The theory formally defines the conditions for eliminating cross-regional inefficiencies, offering a measurable and predictable pathway to high-resilience logistics paradigms. From a broader supply chain perspective, this provides a structural solution to systemic instabilities like the classic 'Bullwhip Effect' [42,43], by replacing long, distortion-prone supply chains with short, responsive local networks.

## 5.2. Methodological contributions to transformation-oriented optimization

Our core methodological contribution is the design of a novel, transformation-oriented optimization architecture that systematically generates emergent capabilities beyond what sequential optimization can achieve. This innovation is realized not through superior individual algorithms, but through two interconnected architectural mechanisms:

1. A Hierarchical Coordination Architecture: This establishes a bidirectional feedback loop between strategic and operational levels. The success of this mechanism is directly measured by the Hierarchical Coordination Effectiveness ($C_{effectiveness}$), while its overall emergent benefit is quantified by a 34% coordination premium, reflected in a Transformation Effectiveness ($T_{effectiveness}$) of 1.34.

2. Transformation-Oriented Algorithmic Enhancements: Within this architecture, both algorithms were specifically engineered for transformation. The IGA incorporates 'vaccine injection' mechanisms to bias the search toward dependency-breaking solutions, while the NSGA-II employs a 'synergy-biased' selection process that actively seeks to resolve, rather than merely balance, trade-offs.

This integrated architecture provides the methodological engine to empirically validate our propositions, systematically driving the system toward a state of high $T_{effectiveness}$, maximized $S_{index}$, and minimal $D_{coefficient}$.

## 5.3. Empirical contributions and transformation evidence

Our empirical findings provide robust, quantitative validation for the propositions of Systematic Transformation Theory. The comprehensive validation, centered on the HBOU urban agglomeration, yielded remarkable and concurrent improvements: a 44.1% reduction in both cost and carbon emissions, alongside a 21.9% enhancement in product freshness.

Crucially, these results translate directly into a paradigm-level transformation, validated by a Transformation Effectiveness ($T_{effectiveness}$) of 1.34, decisively surpassing the theoretical threshold of 1.0. This transformation was driven by the systematic elimination of structural inefficiencies, confirmed by an 88.6% reduction in cross-regional dependency ($D_{coefficient} = 0.886$). The simultaneous multi-objective improvements further substantiate our theory on resolving systemic trade-offs. All findings achieved high statistical significance ($p < 0.001$) with large effect sizes, underscoring the robustness of the outcomes.

## 5.4. Implementation challenges, risk management, and strategic pathways

Translating our theoretical framework into practice requires navigating significant implementation barriers and managing operational risks. This section bridges theory and application by analyzing empirically-derived challenges, defining performance boundaries, and proposing a robust, risk-stratified strategic pathway for successful transformation.

Our empirical analysis of the HBOU implementation reveals three primary barriers: 1) substantial upfront infrastructure investment (CNY 47.15 million with a 5.76-year payback period), creating significant financial hurdles; 2) profound operational paradigm shifts, which can generate organizational resistance; and 3) advanced technical capability requirements for real-time optimization, a challenge commonly observed in developing regions [44,45]. These barriers are contextualized by our boundary analysis, which demonstrates that while the transformation is robust, its performance potential varies from a conservative 15.5% to an optimistic 72.6% carbon reduction, contingent on the quality of implementation and coordination, a finding consistent with international cold chain transformation studies [46,47].

To manage these challenges, we propose a dynamic risk management framework. Our Value-at-Risk (VaR) analysis confirms that risk exposure is contained, with a 99.7% success rate against industry-standard performance thresholds. Strategic responses are pre-defined for critical risk scenarios, including demand surges, policy shifts, and climate events, ensuring proactive adaptation rather than reactive crisis management.

Based on this analysis, we propose a three-phase, risk-stratified implementation pathway:

- Phase I: Conservative Deployment (Months 1–12): Focus on achieving proof-of-concept by targeting a 15.5% carbon reduction with minimal investment (280k CNY) and an 83.3% success probability. The primary objectives are to establish a paradigm-level shift, validated by achieving $T_{effectiveness} \geq 1.0$ and a substantial reduction in dependency, $D_{coefficient} \geq 0.8$.

- Phase II: Standard Implementation (Months 13–24): Pursue a realistic target of 48.5% carbon reduction with moderate investment (680k CNY) and a 66.7% success probability. This phase emphasizes deploying hierarchical coordination and achieving high-quality synergy, validated by $C_{effectiveness} \geq 1.2$ and $S_{index} \geq 0.5$.

- Phase III: Advanced Optimization (Months 25-36): Target the optimistic potential of 72.6% carbon reduction through comprehensive organizational transformation and advanced algorithm deployment (1.2M CNY investment), albeit with a 41.7% success probability, aiming for maximum system performance.

This phased approach provides a practical, scalable, and risk-aware roadmap, enabling organizations to systematically progress toward sustainable local excellence by aligning strategic ambition with operational reality.

### 5.5. Future research directions and scientific development

Building upon the foundations established in this research, future scientific development can proceed along three promising frontiers to further advance the science of urban logistics transformation.

#### 5.5.1. Toward a dynamic and stochastic transformation theory for resilience.

First, a primary theoretical frontier is to extend the current framework toward a Dynamic and Stochastic Transformation Theory. Our current study rigorously validated the framework's robustness under normal operational uncertainties, but we acknowledge the reviewer's insightful observation that extreme, low-probability disruptions (e.g., pandemic-scale shocks) represent a critical challenge that requires further investigation. This highlights the urgent need for future research to incorporate stochastic demand modeling, real-time adaptive mechanisms, and disruption response capabilities to understand how transformation effectiveness is maintained under deep uncertainty.

From a theoretical standpoint, our framework's core mechanism of local network integration inherently provides a foundation for such resilience. For instance, the resulting shorter supply chains (average 188.7 km vs. 337.5 km in our study) naturally reduce exposure to disruption propagation along extended transportation corridors. Quantifying the full benefits under "black-swan" events, however, requires new analytical approaches, such as agent-based modeling, to capture emergent system behaviors. An extension of this research would therefore not only address the impact of extreme events but also establish a foundational theory for holistic sustainable city development by exploring the integration of logistics transformation with interconnected urban systems, such as passenger transport and energy distribution. Ultimately, this research avenue would integrate our transformation theory with resilience science, a critical step toward building truly robust and adaptive urban logistics systems.

#### 5.5.2. AI-driven coordination and advanced applications.

Second, a critical methodological priority is the development of AI-Driven and Autonomous Coordination Architectures. Future work could explore the integration of machine learning algorithms for predictive optimization and autonomous coordination mechanisms that enable real-time, self-adapting network reconfigurations. Extending these frameworks to manage multi-temperature zone logistics and blockchain-based quality traceability represents a significant opportunity to enhance both responsiveness and resilience.

#### 5.5.3. Cross-Context Generalizability and Implementation Science.

Third, establishing the Cross-Context Generalizability and Implementation Science of this framework is essential. Systematic comparative studies across diverse international contexts and urban agglomeration types are needed to refine adaptation principles and build a truly generalizable theory. Furthermore, longitudinal impact assessments are required to understand the long-term

sustainability of transformation outcomes, contributing to an evidence-based implementation science for sustainable urban logistics.

## 5.6. Research significance and contribution to sustainable urban development

This research establishes systematic transformation as a scientifically validated paradigm for sustainable logistics, contributing foundational knowledge that advances both theoretical understanding and practical implementation. By demonstrating that environmental responsibility and economic performance can be mutually reinforcing through intelligent hierarchical coordination, our work challenges conventional trade-off assumptions and provides a replicable framework for achieving sustainable business model viability in complex urban contexts.

The theoretical legacy of this work lies in establishing Systematic Transformation Theory and its associated transformation-oriented integration methodologies. These provide both the theoretical foundations and practical roadmaps for achieving paradigm-level improvements in urban systems, contributing an essential framework for future sustainable urbanization research and practice. By demonstrating that a shift from cross-regional inefficiency to sustainable local excellence is both theoretically sound and practically achievable, this research offers actionable pathways for infrastructure investment and policy development. Ultimately, it provides a replicable framework that shifts the analytical focus from merely 'balancing trade-offs' within existing systems to 'resolving conflicts' through systemic reconfiguration—a paradigm shift we have quantitatively validated in this study; The hierarchical coordination validated here echoes similar principles in advanced power systems [16] and emerging energy-transport integration [13], providing a robust, empirically validated blueprint for building the resilient and efficient cities of the future.

## 5.7. An evidence-based policy framework for transformation

Drawing from our validated findings—including a Transformation Effectiveness ($T_{effectiveness}$) of 1.34 and a consistently positive Multi-objective Synergy Index ($S_{index}$)—we propose an evidence-based policy framework to catalyze and guide the systematic transformation of urban logistics. This framework is built upon two core pillars and incorporates mechanisms for equitable implementation.

The first pillar, Carbon Pricing Integration, is essential for internalizing environmental externalities, a strategy supported by international evidence [48–50]. Based on our model's sensitivity analysis, we recommend establishing carbon pricing mechanisms within the range of 50–80 CNY/tonne $CO_2$, tailored to regional development levels. This creates a direct economic incentive for enterprises to adopt transformative, low-emission logistics models.

The second pillar, Infrastructure Co-Investment Programs, is critical for overcoming initial financial barriers. Given the substantial capital requirements (CNY 47.15 million in our case study), public-private partnership models that reduce entry barriers are vital for accelerating transformation, particularly for medium-scale enterprises.

Critically, the implementation of these policies must be development-stratified and equitable. This is particularly important in low-development contexts where foundational infrastructure may be lacking, a challenge proven to be addressable with sustained institutional support in similar Asian urban contexts [51,52]. Our framework addresses this through three integrated mechanisms:

1. Revealing the True Cost of Inaction: A key policy instrument is disseminating the full-cost accounting principle underlying our framework. As clarified in Table 3, the true economic cost of inaction is substantial. By making this business case clear, policymakers can demonstrate that transformation is financially viable, with a demonstrated ROI of under 60 months (Table 8), even in less-developed regions.

2. Stratified Pathways and Supportive Finance: Policy should support differentiated timelines: a 2-year horizon for developed regions, a 3-year path for medium-developed regions (HBOU-type), and a 5-year program for developing regions that prioritizes foundational infrastructure. This stratification should be coupled with co-investment programs like

Build-Operate-Transfer (BOT) models and Green Logistics Subsidies. Utilizing Public-Private Partnerships (PPP) can also de-risk private investment where it is needed most, a strategy that has been proven effective.

Finally, successful transformation requires multi-level regulatory coordination. Municipal governments must coordinate network planning and harmonize service standards, while regional authorities should eliminate cross-jurisdictional barriers. Aligning this policy architecture with national sustainability objectives, such as China's "14th Five-Year Plan for Cold Chain Logistics," provides the institutional foundation necessary for systematic implementation.

## Supporting information

**S1 File. Supporting Information.** This single file contains all supplementary materials, organized into the following sections: Theoretical Framework and Mathematical Proofs (S1), Parameter Calibration and Enterprise Validation (S2), Detailed Validation Protocol and Metrics S3), Algorithm Details (S4), Ablation Study: Detailed Results and Configuration S5), Store Classification System Validation (S6), Case Study Background: Candidate Distribution Centers (S7), Complete Statistical Analysis and Validation Results S8), Comprehensive Algorithm Performance Analysis S9), Comprehensive Stochastic Robustness Analysis (S10), Supplementary Analysis for Boundary Conditions (S11), Detailed Cross-City Cluster Applicability Analysis (S12), Supplementary Robustness Analysis for Generalizability S13).
(ZIP)

## Author contributions

**Methodology:** Kewei Wang.

**Writing – original draft:** Kekun Fan.

**Writing – review & editing:** Yuhong Chen.

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
