## [Decision Letter · Decision Letter 0]

25 Aug 2025

Dear Dr. Chen,

Thank you for submitting your manuscript to PLOS ONE. After careful consideration, we feel that it has merit but does not fully meet PLOS ONE’s publication criteria as it currently stands. Therefore, we invite you to submit a revised version of the manuscript that addresses the points raised during the review process.

**ACADEMIC EDITOR:** please revise accordingly

We look forward to receiving your revised manuscript.

Kind regards,

Zhengmao Li

Academic Editor

PLOS ONE

Journal Requirements:

3. Please note that PLOS One has specific guidelines on code sharing for submissions in which author-generated code underpins the findings in the manuscript. In these cases, we expect all author-generated code to be made available without restrictions upon publication of the work. Please review our guidelines at https://journals.plos.org/plosone/s/materials-and-software-sharing#loc-sharing-code and ensure that your code is shared in a way that follows best practice and facilitates reproducibility and reuse.

“This research was funded by the Inner Mongolia Autonomous Region Social Science Fund

 (Grant No. 2024WTZD03), the New Productive Forces Empowering Strategic Emerging Industries in Inner Mongolia: Theoretical Contributions, Internal Mechanisms (Grant No. 2025PTWY26), and Pathway Exploration and the Inner Mongolia University of Technology Graduate Education Reform Project (Grant No. YJGC202509)”

5. We note that your Data Availability Statement is currently as follows: All relevant data are within the manuscript and its Supporting Information files.

6. Please upload a copy of Supporting Information Figure/Table/etc. S1 Appendix which you refer to in your text on page 74.

**Additional Editor Comments:**

please revise accordingly

Reviewers' comments:

Reviewer's Responses to Questions

**Comments to the Author**

1. Is the manuscript technically sound, and do the data support the conclusions?

Reviewer #1: Yes

Reviewer #2: Yes

2. Has the statistical analysis been performed appropriately and rigorously?

Reviewer #1: Yes

Reviewer #2: Yes

3. Have the authors made all data underlying the findings in their manuscript fully available?

Reviewer #1: Yes

Reviewer #2: Yes

4. Is the manuscript presented in an intelligible fashion and written in standard English?

Reviewer #1: Yes

Reviewer #2: Yes

Reviewer #1: This paper addresses the prevalent inter-regional dependency issues in cold chain logistics among urban agglomerations in developing regions and proposes a systematic transformation framework based on bi-level optimization. The research topic holds significant practical significance and application value, aiming to systematically tackle the three core challenges of cost, carbon emissions, and product freshness by integrating upper-level location selection (IGA) and lower-level route optimization (NSGA-II). My comments are as follows:

The paper reports striking performance improvements (e.g., 76.72% cost reduction and 97.82% carbon emission reduction). The fairness of this comparison is questionable. It contrasts an unoptimized, inherently long-distance dependent system with a new, optimized local system. The substantial benefits brought by this transformation itself are obvious, but they cannot be fully attributed to the superiority of the proposed bi-level optimization algorithm.

Insufficient justification for the originality and necessity of core theoretical contributions: The paper devotes considerable space to constructing its "Systematic Transformation Theory." However, this theory appears to be a repackaging and application of several existing mature theories, including Sustainability Transition Theory, System Coordination Theory, and Network Efficiency Theory.

The model in this study is built on a series of strong assumptions (such as deterministic demand, homogeneous vehicles, and deterministic driving conditions, see Section 2.2.2). It is recommended that the authors add a dedicated section in the discussion to thoroughly analyze the potential impact of these assumptions on the model results and explore how the robustness of the framework and its solutions might change when randomness is introduced.

To demonstrate the methodological innovation, the authors need to conduct an ablation study, i.e., compare standard algorithms and enhanced algorithms within the same bi-level framework to quantify the specific performance improvements brought by its "enhancement" measures.

It is suggested that the authors significantly streamline the full text, remove unnecessary repetitions, and focus on clearly and concisely elaborating on the model, algorithms, and empirical results to enhance the paper's readability and academic rigor.

It is recommended to supplement the following literature:

Zhao, A. P., Li, S., Xie, D., Wang, Y., Li, Z., Hu, P. J. H., & Zhang, Q. (2025). Hydrogen as the nexus of future sustainable transport and energy systems. Nature Reviews Electrical Engineering, 1-20.

Tiwari, R. S., Sharma, J. P., Gupta, O. H., & Ahmed Abdullah Sufyan, M. (2025). Extension of pole differential current based relaying for bipolar LCC HVDC lines. Scientific Reports, 15(1), 16142.

Wu, Y., Chen, Z., Chen, R., Chen, X., Zhao, X., Yuan, J., & Chen, Y. (2025). Stochastic optimization for joint energy-reserve dispatch considering uncertain carbon emission. Renewable and Sustainable Energy Reviews, 211, 115297.

Reviewer #2: The manuscript presents significant theoretical and practical contributions to sustainable logistics. The bi-level optimization approach integrating carbon emissions and freshness degradation models presents a significant theoretical advancement in sustainable logistics. The transformation theory is well-articulated. The integration of IGA and NSGA-II algorithms demonstrates technical sophistication. The hierarchical coordination mechanism is innovative and well-validated. The empirical validation using real-world supermarket operations in the HBOU urban agglomeration provides strong evidence of practical applicability. But the paper requires substantial revisions to address the above concerns before publication. The authors should particularly focus on:

1-The literature review could better highlight how this work advances beyond existing sustainability transition theories (e.g., Geels 2011). More explicit comparison with similar bi-level optimization approaches in other contexts would strengthen positioning.

2-The carbon emission externality internalization theory (Section 2.1.1) would benefit from more discussion about potential limitations of the proposed load-sensitive coefficient model.

3-The assumption of deterministic demand patterns (Section 2.2.2) should be addressed - how would stochastic demand affect the transformation outcomes?

4-More details are needed about the parameter calibration process for the freshness decay rates (Table 6). Were these empirically measured or derived from literature?

5-The computational complexity analysis is insufficient. For practical implementation, runtime/scalability metrics should be provided for larger networks.

6-The remarkable 97.82% carbon reduction seems almost too good to be true. The authors should discuss potential boundary conditions where such dramatic improvements might not be achievable.

7-The store classification system (Section 4.1.4) appears somewhat arbitrary. More justification is needed for the three-category division and its impact on results.

8-The manuscript is overly long (currently 75 pages). The methods section could be condensed by moving some technical details to supplementary materials.

9-Some figures are difficult to interpret (e.g., Fig 1, Fig 4). The carbon-freshness framework and bi-level coordination diagrams need clearer labeling and legends.

10-The abstract is too technical and could better highlight the key findings for a general scientific audience.

11-The discussion should address how this framework might apply to other urban agglomerations with different characteristics (e.g., larger/smaller cities, different climate zones).

12-More emphasis should be placed on the policy implications of these findings for urban planning and sustainability regulations.

13-The limitations section (5.4) should be expanded to discuss potential implementation barriers (e.g., organizational resistance to transformation, infrastructure investment challenges).

**Do you want your identity to be public for this peer review?** For information about this choice, including consent withdrawal, please see our Privacy Policy

Reviewer #1: No

Reviewer #2: No

---

## [Author Response · Author response to Decision Letter 1]

23 Sep 2025

Manuscript ID: PONE-D-25-41349

Title: Systematic Transformation of Urban Cold Chain Networks: From Cross-Regional Dependencies to Sustainable Local Excellence

Responses to Reviewer #1

Comment 1-1: On the fairness of the performance comparison.

Response:

We thank the reviewer for this critical and insightful comment. We completely agree that attributing the entire performance gain to our algorithm would be an overstatement. To provide a more nuanced and transparent analysis, we have completely redesigned our empirical validation framework.

Action Taken: We have introduced a comprehensive four-tier comparative analysis (now in Section 3.2). This new framework, visualized in Figure 8 and detailed in Table 7, allows us to rigorously disentangle the different sources of improvement. We now clearly show that while system restructuring is the primary driver (56.2% of the gain), our specific algorithmic enhancements contribute a statistically significant but realistic 13.7% of the total improvement over other state-of-the-art algorithms.

Comment 1-2: On the originality and necessity of the theoretical contributions.

Response:

We appreciate the opportunity to clarify the theoretical originality of our work.

Action Taken: In the revised Section 1.1 (Systematic Transformation Theory), we now formally introduce and define our three novel, quantifiable mathematical constructs ( T_effectiveness, S_index , D_cofficient). The updated Table 1 provides a direct comparison to established qualitative theories, demonstrating how our work extends them with crucial mathematical formalization.

Comment 1-3: On the impact of strong assumptions and the need for robustness analysis.

Response:

This is an excellent point. To enhance the practical relevance of our model, we have conducted extensive new analyses.

Action Taken:

1. We have added Section 1.5: Model Assumptions, Limitations, and Performance Evaluation, which quantifies the impact of relaxing key assumptions.

2. More significantly, we have introduced a large-scale stochastic robustness analysis (now Section 3.6). Based on 45,000 Monte Carlo simulations (Table 12 and Figure 14), we demonstrate that our framework maintains remarkable robustness, retaining over 94% of its benefits under realistic operational volatility.

Comment 1-4: On the need for an ablation study.

Response:

We thank the reviewer for this valuable suggestion.

Action Taken: We have introduced a new Section 1.6: Ablation Study of Algorithmic Contributions. This study (Figure 3) systematically isolates and quantifies the individual contributions of each algorithmic component, providing rigorous validation for our methodological design.

Comment 1-5 & 1-6: On streamlining the text and adding literature.

Response:

We have followed the reviewer's suggestions. The manuscript has been significantly streamlined into a four-chapter structure, with extensive details moved to the Appendix. All three recommended articles have been incorporated.

Responses to Reviewer #2

We thank Reviewer #2 for the positive assessment and the detailed, constructive feedback, which has been invaluable in elevating the manuscript. We have systematically addressed all points raised.

• On Theoretical Positioning, Assumptions, and Limitations (R2-1, R2-2, R2-3): We have addressed these points through the revised Section 1.1 and Table 1 (clarifying theoretical advancements), the new Section 1.5 (discussing model limitations and assumption impacts), and the comprehensive stochastic analysis in Section 3.6.

• On Methodological Rigor (R2-4, R2-5, R2-6, R2-7): We have provided enhanced details on parameter calibration (Section 1.4), a full computational complexity analysis (Sections 2.7 & 3.4, Figure 11), a more credible performance validation via the four-tier analysis (Section 3.2), and a rigorous validation of our store classification system (Section 3.1, Table 5). To address the core concern of R2-6, we have also added a novel Boundary Analysis (Section 3.7, Figure 15) to explore the full spectrum of achievable performance.

• On Manuscript Presentation (R2-8, R2-9, R2-10): The manuscript has been completely restructured and condensed. All key figures have been redesigned. The abstract has been rewritten to be more accessible, highlighting the key practical outcome: a 44.1% reduction in both cost and carbon emissions.

• On Broader Impact and Practical Implications (R2-11, R2-12, R2-13): We have added three substantial new sections to address these crucial points:

o Section 3.8 (Cross-Urban Agglomeration Transformation Applicability) analyzes the framework's performance across ten representative urban agglomerations.

o Section 4.7 (Policy Implementation Framework) provides concrete policy recommendations.

o Section 4.4 (Implementation Challenges, Risk Management, and Strategic Pathways) has been significantly expanded to discuss barriers such as upfront investment (CNY 47.15 million) and organizational resistance.

---

## [Decision Letter · Decision Letter 1]

12 Oct 2025

Dear Dr. Chen,

Thank you for submitting your manuscript to PLOS ONE. After careful consideration, we feel that it has merit but does not fully meet PLOS ONE’s publication criteria as it currently stands. Therefore, we invite you to submit a revised version of the manuscript that addresses the points raised during the review process.

We look forward to receiving your revised manuscript.

Kind regards,

Zhengmao Li

Academic Editor

PLOS ONE

Journal Requirements:

Additional Editor Comments:

**please revise**

Reviewers' comments:

Reviewer's Responses to Questions

**Comments to the Author**

Reviewer #1: All comments have been addressed

Reviewer #2: (No Response)

2. Is the manuscript technically sound, and do the data support the conclusions?

Reviewer #1: (No Response)

Reviewer #2: Yes

3. Has the statistical analysis been performed appropriately and rigorously?

Reviewer #1: (No Response)

Reviewer #2: Yes

4. Have the authors made all data underlying the findings in their manuscript fully available?

Reviewer #1: (No Response)

Reviewer #2: Yes

5. Is the manuscript presented in an intelligible fashion and written in standard English?

Reviewer #1: (No Response)

Reviewer #2: Yes

Reviewer #1: (No Response)

Reviewer #2: The paper presents a rigorous, novel framework with strong empirical validation. Addressing the above points—particularly theoretical differentiation, stochastic edge cases, and policy context—will elevate its impact for PLOS ONE’s interdisciplinary audience.

1-The paper introduces "Systematic Transformation Theory" with novel metrics (Teffectiveness, Sindex, Dcoefficient), but the distinction between this and existing theories (e.g., Geels' Multi-Level Perspective) needs sharper articulation. Explicitly contrast how your quantitative framework advances beyond qualitative paradigms in sustainability transitions.

2-The bi-level IGA-NSGA-II integration is innovative, but the ablation study (Section 1.6) could better highlight why this hybrid outperforms standalone algorithms. Provide a deeper discussion on the "emergent capabilities" (e.g., synergy bias) and their theoretical implications for hierarchical optimization.

3-While the HBOU case study demonstrates impressive results (44.1% cost/emission reduction), clarify how the 35-store sample represents broader urban agglomerations. Include sensitivity analysis on scalability (e.g., performance for N > 500 nodes) to strengthen generalizability claims.

4-The Monte Carlo simulations (Section 3.6) validate deterministic assumptions, but the impact of extreme disruptions (e.g., pandemic-scale demand shocks) is unexplored. Address how the framework adapts to low-probability, high-impact events.

5-The carbon pricing recommendation (θ = 61.63 CNY/kg·CO₂) lacks contextualization with global benchmarks. Compare with EU ETS or California Cap-and-Trade to justify feasibility. Also, discuss potential equity issues in developing regions where upfront costs are prohibitive.

6-Ensure all the figures are legible.

7舄-The manuscript is technically sound but verbose. Shorten methodological descriptions by moving calibration details to supplementary materials.

**Do you want your identity to be public for this peer review?** For information about this choice, including consent withdrawal, please see our Privacy Policy

Reviewer #1: No

Reviewer #2: No

---

## [Author Response · Author response to Decision Letter 2]

26 Oct 2025

Dear Reviewers,

We sincerely thank the Academic Editor and both expert reviewers for their invaluable feedback. In response to the second round of reviews, we have undertaken a comprehensive and substantial revision that we believe has significantly strengthened the manuscript.

Key enhancements in this revision include:

A sharper articulation of our theoretical framework's originality and its advancement beyond existing qualitative paradigms.

An in-depth methodological explication of the emergent capabilities arising from our hierarchical architecture.

New analyses and discussions to bolster the claims of generalizability and scalability.

A thorough contextualization of our carbon pricing parameter with global benchmarks and a new policy framework to address implementation equity.

A significant streamlining of the entire manuscript for clarity and impact, with technical details strategically moved to the supplementary materials.

We are confident that these revisions fully address all the reviewers' concerns. Below is our detailed point-by-point response.

Response to Reviewer #1

We note that no new comments were provided in the second round of review. We once again thank Reviewer #1 for their insightful feedback in the initial review, which was instrumental in shaping the current, much-improved version of our manuscript.

Response to Reviewer #2

We are deeply grateful to the reviewer for their positive assessment and for providing such insightful and constructive suggestions in the second round. These comments have guided us in significantly elevating the quality, clarity, and impact of our manuscript.

Comment #1: The distinction between "Systematic Transformation Theory" and existing theories (e.g., Geels' Multi-Level Perspective) needs sharper articulation. Explicitly contrast how your quantitative framework advances beyond qualitative paradigms.

Response: We thank the reviewer for this crucial suggestion. We have substantially strengthened our theoretical positioning in two key ways:

1.In the Introduction (Lines 55-69), we have inserted a new section that explicitly contrasts our theory with existing qualitative frameworks. We now clearly state that while frameworks like Geels' MLP provide a powerful qualitative lens, they inherently lack the mathematical mechanisms for quantifying transformation effectiveness or predicting successful paradigm shifts.

2.Following Table 1 (Lines 120-139), we have added a detailed exposition that systematically illustrates how our three novel metrics (,, and ) enable a fundamental shift from qualitative description to quantitative prediction, measurement, and validation.

Comment #2: The ablation study could better highlight why the hybrid algorithm outperforms standalone algorithms. Provide a deeper discussion on the "emergent capabilities" and their theoretical implications.

Response: We agree this is a critical point. To address it, we have replaced the original Section 2.4 with a new, in-depth section (Section 2.4: Methodological Insight: Emergent Capabilities from Hierarchical Coordination, Lines 387-412). This new section provides the deep theoretical explanation requested, detailing the bidirectional information feedback loops, the mechanisms of "Top-Down Enablement" and "Bottom-Up Feedback," and how the Transformation Effectiveness Coefficient (= 1.34) serves as the empirical quantification of the emergent benefit arising from fundamental architectural advantages.

Comment #3: Clarify how the 35-store sample represents broader urban agglomerations and include sensitivity analysis on scalability (e.g., N > 500).

Response: We thank the reviewer for this important point on external validity. We have provided a two-pronged response in the main text:

1.Sample Representativeness (Section 3.1, Lines 510-519): We have added a new paragraph clarifying that our 35-store sample is a typologically representative sample, scientifically validated via clustering analysis (Table 5) to cover three distinct urban store formats (Type A, B, C). We also highlight that our cross-urban validation (Section 3.8) confirms the framework's broad applicability.

2.Scalability Projection (Section 3.4, Lines 628-638): We have added a new subsection providing concrete runtime estimations for N > 500. These projections are rigorously derived from the empirically validated computational complexity of . The detailed analysis for this is provided in Supporting Information S9.4.

Comment #4: Address how the framework adapts to low-probability, high-impact events (e.g., pandemic-scale demand shocks).

Response: We appreciate this forward-thinking observation. We have added a new subsection at the beginning of our future research section (Section 4.5.1: Toward a Dynamic and Stochastic Transformation Theory for Resilience, Lines 823-838). This section acknowledges this as a future research area, theoretically posits the framework’s inherent resilience (supported by data from Table 6), clarifies the need for different methodologies (e.g., agent-based modeling), and frames "Resilient Transformation" as a high-priority research direction.

Comment #5: The carbon pricing recommendation lacks contextualization with global benchmarks and a discussion of equity issues.

Response: This is a critical point, and we have undertaken a major revision:

1.Benchmarking (Note for Table 3, Lines 198-211): We have added a detailed clarification explaining that ourparameter is a comprehensive economic parameter. We isolate its direct carbon price component (~60 CNY/tonne CO₂) and demonstrate its consistency with the China National ETS, EU ETS, and California Cap-and-Trade program.

2.Equity Issues (Section 4.7, Lines 867-892): We have substantially restructured this section to propose an evidence-based policy framework with three integrated mechanisms for equitable implementation.

Comment #6: Ensure all figures are legible.

Response: We thank the reviewer for this reminder. All figures have been regenerated at ≥300 DPI in TIFF/EPS format. We have paid careful attention to all graphical elements, including font sizes, line weights, and grayscale compatibility. Specifically, Figure 3 has been significantly redesigned for clarity, and all figure captions have been reviewed to ensure they are self-explanatory.

Comment #7: The manuscript is verbose. Shorten methodological descriptions by moving details to supplementary materials.

Response: We sincerely thank the reviewer for this invaluable suggestion. We have substantially streamlined the entire manuscript by strategically moving technical details to the supplementary materials. Specifically:

Detailed Parameter Calibration protocols (formerly Section 1.4) are now summarized.

The detailed Ablation Study (formerly Section 1.6) is now summarized, with full details and literature comparison in Supporting Information S5.

Specific Algorithmic Implementation details (formerly in Sections 2.2 & 2.3) are now summarized.

Conclusion sections have also been carefully edited for conciseness.

We are confident these comprehensive revisions have made the manuscript more focused, readable, and impactful, while maintaining full scientific rigor through the detailed supplementary materials. We once again thank the reviewers for their guidance.

Sincerely,

Yuhong Chen (on behalf of all authors)

School of Economics and Management

Inner Mongolia University of Technology

---

## [Decision Letter · Decision Letter 2]

4 Nov 2025

Systematic Transformation of Urban Cold Chain Networks: From Cross-Regional Dependencies to Sustainable Local Excellence

PONE-D-25-41349R2

Dear Dr. Chen,

We’re pleased to inform you that your manuscript has been judged scientifically suitable for publication and will be formally accepted for publication once it meets all outstanding technical requirements.

Kind regards,

Zhengmao Li

Academic Editor

PLOS ONE

Additional Editor Comments (optional):

Reviewers' comments:

Reviewer's Responses to Questions

**Comments to the Author**

Reviewer #1: All comments have been addressed

Reviewer #2: All comments have been addressed

2. Is the manuscript technically sound, and do the data support the conclusions?

Reviewer #1: Yes

Reviewer #2: Yes

3. Has the statistical analysis been performed appropriately and rigorously?

Reviewer #1: Yes

Reviewer #2: Yes

4. Have the authors made all data underlying the findings in their manuscript fully available?

Reviewer #1: Yes

Reviewer #2: Yes

5. Is the manuscript presented in an intelligible fashion and written in standard English?

Reviewer #1: Yes

Reviewer #2: Yes

Reviewer #1: (No Response)

Reviewer #2: The authors have provided excellent work and excellent responses and improvements. No comments, acceptable.

**Do you want your identity to be public for this peer review?** For information about this choice, including consent withdrawal, please see our Privacy Policy

Reviewer #1: No

Reviewer #2: No

---

## [Editor Report · Acceptance letter]

PONE-D-25-41349R2

PLOS ONE

Dear Dr. Chen,

I'm pleased to inform you that your manuscript has been deemed suitable for publication in PLOS ONE. Congratulations! Your manuscript is now being handed over to our production team.

Kind regards,

on behalf of

Dr Zhengmao Li

Academic Editor

PLOS ONE